# Fair GLASSO: Estimating Fair Graphical Models with Unbiased Statistical Behavior

**Madeline Navarro**
Rice University
nav@rice.edu

**Samuel Rey**
King Juan Carlos University
samuel.rey.escudero@urjc.es

**Andrei Buciulea**
King Juan Carlos University
andrei.buciulea@urjc.es

**Antonio G. Marques**
King Juan Carlos University
antonio.garcia.marques@urjc.es

**Santiago Segarra**
Rice University
segarra@rice.edu

## Abstract

We propose estimating *Gaussian graphical models (GGMs)* that are *fair with respect to sensitive nodal attributes*. Many real-world models exhibit unfair discriminatory behavior due to biases in data. Such discrimination is known to be exacerbated when data is equipped with pairwise relationships encoded in a graph. Additionally, the effect of biased data on graphical models is largely underexplored. We thus introduce fairness for graphical models in the form of two bias metrics to promote balance in statistical similarities across nodal groups with different sensitive attributes. Leveraging these metrics, we present Fair GLASSO, a regularized graphical lasso approach to obtain sparse Gaussian precision matrices with unbiased statistical dependencies across groups. We also propose an efficient proximal gradient algorithm to obtain the estimates. Theoretically, we express the tradeoff between fair and accurate estimated precision matrices. Critically, this includes demonstrating when accuracy can be preserved in the presence of a fairness regularizer. On top of this, we study the complexity of Fair GLASSO and demonstrate that our algorithm enjoys a fast convergence rate. Our empirical validation includes synthetic and real-world simulations that illustrate the value and effectiveness of our proposed optimization problem and iterative algorithm.

## 1 Introduction

Data analysis frequently requires estimating complex dyadic relationships, which can be conveniently encoded in graphical representations such as Gaussian graphical models (GGMs) [1–3]. Myriad real-world applications model structure in data by obtaining graphs from observations, in fields including neuroscience, genomics, finance, and more [4–6]. However, it is known that real-world data can encode historical biases which models ought not to consider, such as discriminatory biases against sensitive populations [7, 8]. For example, social networks often exhibit preferential relationships that may unfairly discriminate against sensitive communities [9–11]. Moreover, the use of unfair graphs for downstream tasks is known to exacerbate existing biases [12–14]. While accurate graphical representations are critical for applications and analyses, the propagation of undesirable bias in graph data necessitates learning models that balance both fairness and accuracy.

The long-standing popularity of GGMs for several applications, many of them high-stakes, warrants care in how we estimate them from potentially biased data. However, there is no formal definition of fairness for graphical models, and existing definitions for graph-based machine learning may not be applicable for obtaining fair statistical relationships. Indeed, while fairness for graph data has recently received copious attention, the study of biased graphs in statistics and graph signal processing (GSP)

38th Conference on Neural Information Processing Systems (NeurIPS 2024).

is only beginning [15–17]. Furthermore, previous works primarily consider fairness for downstream tasks, while few attempt to learn unbiased graphs from data [16–18]. We thus arrive at two vital questions. First, *what does it mean for a graphical model to be fair?* We aim to compare such a notion to existing definitions of fairness on graphs. Second, *how can we obtain GGMs that are fair in the presence of biased data?* To address these questions, we consider estimation of fair GGMs from biased observations, where nodes belong to groups corresponding to different sensitive attributes.

We propose an optimization framework to obtain fair GGMs from potentially biased data, where statistical dependencies between nodes show no preferences for particular groups. We first define fairness for graphical models by introducing two bias metrics that measure similarities in statistical behavior between pairs of groups. Our metrics are simple and convex, yet they intuitively capture biases in terms of conditional dependence. We then propose *Fair GLASSO*, a penalized maximum likelihood estimator using our bias metrics as regularizers, which aims to obtain sparse Gaussian precision matrices that optimally extract structural information from observed data while promoting fairer statistical behavior across node groups. We summarize our contributions as follows:

- *We formally define fairness for graphical models* via two bias metrics, one balancing statistical dependencies evenly across all groups and a stronger alternative requiring each node to be balanced across all groups. We relate our definition to other notions of fairness on graphs, where ours is specific to graphs encoding conditional dependence structures, which in turn allows greater interpretability and more detailed statistical analysis.
- We present *Fair GLASSO*, a penalized maximum likelihood estimator for sparse Gaussian precision matrices that are unbiased according to any measure of graphical fairness, which we demonstrate with our proposed bias metrics. We theoretically demonstrate that our approach yields a tradeoff between fairness and accuracy, which depends on the bias in the underlying graph.
- The convexity of Fair GLASSO under our proposed metrics allows us to propose *an efficient iterative method* based on proximal gradient descent. We show that our algorithm enjoys iterations of moderate complexity and provable convergence.
- We evaluate Fair GLASSO on both *synthetic and real-world datasets*. The former provides empirical validation of the efficiency of our algorithm and the existence of the fairness-accuracy tradeoff. The latter shows the myriad real-world applications for which we can reliably obtain graphical representations from data while also balancing statistical behavior across sensitive groups.

### 1.1 Notation

For any positive integer $p \in \mathbb{N}$, we let $[p] := \{1, 2, \ldots, p\}$. For a matrix $\mathbf{X} \in \mathbb{R}^{p \times p}$ and a set of indices $\mathcal{C} \in [p]^2$, we let $\mathbf{X}_{\mathcal{C}}$ denote a masking operation on $\mathbf{X}$ permitting non-zero entries only at indices in $\mathcal{C}$. If we define $\mathcal{D} := \{p(i-1) + i\}_{i=1}^{p}$, then $\mathbf{X}_{\mathcal{D}}$ is a diagonal matrix containing the diagonal entries of $\mathbf{X}$. For $\bar{\mathcal{D}} := [p]^2 \backslash \mathcal{D}$ denoting the complement of the set $\mathcal{D}$, $\mathbf{X}_{\bar{\mathcal{D}}} = \mathbf{X} - \mathbf{X}_{\mathcal{D}}$ contains non-zero values of $\mathbf{X}$ only in its off-diagonal entries. We also let $\text{vec}(\mathbf{X}) \in \mathbb{R}^{p^2}$ denote the vertical concatenation of the columns of $\mathbf{X}$. The smallest and largest eigenvalues of a matrix $\mathbf{X}$ are represented respectively by $\lambda_{\min}(\mathbf{X})$ and $\lambda_{\max}(\mathbf{X})$.

## 2 Fair Gaussian Graphical Models

GGMs succinctly model pairwise relationships in multivariate Gaussian distributions through intuitive graphical representations. We denote undirected graphs by $\mathcal{G} = (\mathcal{V}, \mathcal{E}, \mathbf{W})$, where $\mathcal{V} = [p]$ is the set of $p$ nodes and $\mathcal{E} \subseteq \mathcal{V} \times \mathcal{V}$ the set of edges connecting pairs of nodes. For graphs with weighted edges, $\mathbf{W} \in \mathbb{R}^{p \times p}$ encodes the topological structure of $\mathcal{G}$ such that $W_{ij} \neq 0$ if and only if $(i, j) \in \mathcal{E}$, that is, there is an edge in $\mathcal{E}$ connecting nodes $i$ and $j$ with weight $W_{ij}$. Let $\boldsymbol{x} \in \mathbb{R}^p$ be a random vector following a zero-mean Gaussian distribution with positive definite covariance matrix $\boldsymbol{\Sigma}_0 \succ 0$, that is, $\boldsymbol{x} \sim \mathcal{N}(\mathbf{0}, \boldsymbol{\Sigma}_0)$. The precision matrix $\boldsymbol{\Theta}_0 = \boldsymbol{\Sigma}_0^{-1}$ completely describes the conditional dependence structure among the variables in $\boldsymbol{x}$. In particular, for any distinct pair $i, j \in [p]$, variables $x_i$ and $x_j$ are conditionally independent if and only if $[\boldsymbol{\Theta}_0]_{ij} = 0$ [1, 2]. This Markovian property yields a graphical representation, where off-diagonal entries of $\boldsymbol{\Theta}$ encode the weighted edges of a graph $\mathcal{G}$ connecting conditionally dependent variables. In this work, we aim to obtain the structure encoded in $\boldsymbol{\Theta}_0$ using observations sampled from $\mathcal{N}(\mathbf{0}, \boldsymbol{\Sigma}_0)$.

When considering group fairness, we associate each variable in $\boldsymbol{x}$ with one of $g$ groups that partition the variables according to a sensitive attribute [20, 21]. We represent group membership by the

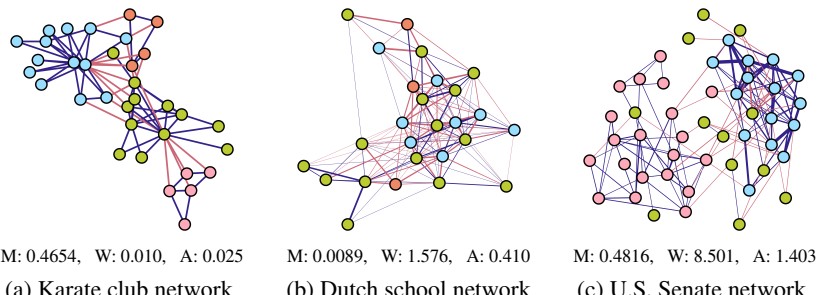

| M: 0.4654, W: 0.010, A: 0.025 | M: 0.0089, W: 1.576, A: 0.410 | M: 0.4816, W: 8.501, A: 1.403 |
|:---:|:---:|:---:|
| (a) Karate club network | (b) Dutch school network | (c) U.S. Senate network |

Figure 1: Three real-world networks with node groups denoted by color. Within-group edges are in blue and across-group edges in red, while edge widths correspond to edge weight magnitudes. For each network, we present ("M") the modularity of the graphs with respect to group membership [19], ("W") the ratio of positive to negative estimated partial correlations for within-group edges, and ("A") an analogous ratio for across-group edges. Networks in (a) and (c) show high group-wise modularity, while (b) and (c) show significant preferences for positive correlations in the same group.

indicator matrix $\mathbf{Z} = [\mathbf{z}_1, \ldots, \mathbf{z}_g] \in \{0,1\}^{p \times g}$, where $Z_{ia} = 1$ if and only if variable $i$ belongs to group $a \in [g]$, otherwise $Z_{ia} = 0$. Group sizes are denoted by $p_a = \sum_{i=1}^{p} Z_{ia}$ for every $a \in [g]$. We also assume that groups are non-overlapping, thus $p = \sum_{a=1}^{g} p_a$. GGMs may possess biases when a group of variables behaves significantly more or less similarly to particular groups. Indeed, individuals within the same political party tend to vote similarly [18]. In this case, entries of $\mathbf{\Theta}_0$ corresponding to pairs of voters of the same party will likely be positive and larger in magnitude.

We empirically demonstrate this phenomenon in Figure 1 for multiple real-world networks. Figures 1a and b show two social network examples, Zachary's karate club network [22, 23] and the Dutch school network [24], where nodes represent individuals and edges connect them by their relationships. Figure 1c is a political network that connects U.S. senators if their voting patterns exhibit correlated behavior [18]. For each network, we present both their modularity with respect to group membership [19] and a comparison of their approximate partial correlations within and across nodal groups [25, 26], with both metrics defined in Appendix G. Figures 1a and c show higher group-wise modularity, in line with the ubiquitous preference for within-group connections in real-world networks [9, 10]. However, observe that Figures 1b and c exhibit a clear preference for positive correlations between nodes in the same group. Despite its presence in real-world interconnected data, this type of discriminatory behavior is typically not addressed in existing works [27]. These examples inspire us to develop a definition of fairness for graphical models that accounts for both *correlation bias*, when behavior is highly correlated between certain groups, and *connectivity bias*, when connections are denser or sparser across certain group pairs.

## 2.1 Bias Metric for Fair Graphical Models

To address biases in both connectivity and correlations, we propose a definition of group fairness for graphical models. We consider the popular notion of demographic parity (DP), the primary choice for fairness on graphs [20]; however, other definitions of group fairness can be similarly adapted [21]. DP requires that outcomes be agnostic to sensitive attributes [20, 28]. In our case, we require estimation of graphical models with unbiased edge selection with respect to nodal groups. Thus, we present the following definition of dyadic DP for graphical models.

**Definition 1** *For a graphical model with the matrix* $\mathbf{\Theta}_0 \in \mathbb{R}^{p \times p}$ *encoding the underlying conditional dependence structure, we consider DP to be satisfied if*

$$\mathbb{P}[[\mathbf{\Theta}_0]_{ij}|Z_{ia} = 1, Z_{ja} = 1] = \mathbb{P}[[\mathbf{\Theta}_0]_{ij}|Z_{ia} = 1, Z_{jb} = 1] \quad \forall\, a, b \in [g]. \tag{1}$$

Note that the distribution need not be Gaussian for this definition. Intuitively, our graphical model DP requires that groups have evenly balanced connections across groups and do not behave significantly more or less similarly to certain groups. Crucially, Definition 1 accounts for both forms of bias showcased in Figure 1, *connectivity* bias in the support of $\mathbf{\Theta}_0$ and *correlation* bias in the signs of entries in $\mathbf{\Theta}_0$. Not only is this definition appropriately tailored to fairness for graphical models, but it

also provides flexibility in how we address biases. We may adjust similarities in behavior without changing the topology of the graph [14, 29], but conversely, we may alter connections if correlations cannot be changed [30]. Finally, note that works for fairness on graphs with weighted or signed edges are rare [27], and to the best of our knowledge no previous work has specified fairness for graphical models that encode conditional dependencies.

We consider a graphical model unfair when there is a gap in DP, that is, when (1) does not hold. In practice, we measure biases in GGMs by approximating the DP gap. For this purpose, we propose the following bias metric inspired by [18],

$$H(\boldsymbol{\Theta}) := \frac{1}{g^2 - g} \sum_{a=1}^{g} \sum_{b \neq a} \left( \frac{\mathbf{z}_a^\top \boldsymbol{\Theta}_{\bar{\mathcal{D}}} \mathbf{z}_a}{p_a^2 - p_a} - \frac{\mathbf{z}_a^\top \boldsymbol{\Theta}_{\bar{\mathcal{D}}} \mathbf{z}_b}{p_a p_b} \right)^2. \tag{2}$$

Each term in (2) compares the average within-group edge weight and the average across-group edge weight for every distinct group pair. Thus, $H(\boldsymbol{\Theta})$ will increase if variables belonging to two groups exhibit either significantly denser or sparser connections or significantly stronger or weaker correlations. As we aim to balance statistical behavior across groups, we consider obtaining precision matrices $\boldsymbol{\Theta}$ that balance data fidelity with small values of $H(\boldsymbol{\Theta})$. The main difference between $H(\boldsymbol{\Theta})$ and the related metric in [18] lies in the use of squared summands. While subtle, this modification tends to yield fairer outcomes, as group pairs are balanced overall as opposed to the metric in [18], which may favor balancing some pairs of groups over others.

In addition to (2), we also propose a stronger alternative metric for node-wise fairness across groups,

$$H_{\text{node}}(\boldsymbol{\Theta}) := \frac{1}{pg} \sum_{i=1}^{p} \sum_{a=1}^{g} \left( \frac{1}{g-1} \sum_{b \neq a} \frac{[\boldsymbol{\Theta}_{\bar{\mathcal{D}}} \mathbf{z}_a]_i}{p_a} - \frac{[\boldsymbol{\Theta}_{\bar{\mathcal{D}}} \mathbf{z}_b]_i}{p_b} \right)^2, \tag{3}$$

which is zero if and only if every variable is completely balanced across groups in terms of connections or correlation. This stronger metric is inspired by [15, 18], also modified by squaring summands as for $H(\boldsymbol{\Theta})$. As an alternative interpretation, observe that $H_{\text{node}}(\boldsymbol{\Theta})$ increases when the correlation between the group of a variable $i$ and the $i$-th column of $\boldsymbol{\Theta}$ increases. This node-wise penalty is stronger than $H(\boldsymbol{\Theta})$, as we require that not only pairs of groups exhibit no preference in statistical similarities but also each node must show no preference for connecting to certain groups.

For graph-based works, the predominant choice of bias metric is DP (see related works in Appendix A). Thus, we approach the nascent task of graphical model estimation with a familiar bias metric to verify our approach with established measurements. However, our formulation is suited to others such as equalized odds (EO), defined in Appendix I. While both DP and EO are popular fairness definitions, we cannot compute EO for the true precision matrix since it is conditioned on the ground truth connections. For this reason, we emphasize DP for group fairness since a measure of bias in the true precision matrix is critical to our theoretical interpretation of the fairness-accuracy tradeoff.

## 3 Fair GLASSO

### 3.1 Graphical Lasso for Fair GGMs

We apply our proposed metrics in (2) and (3) to estimate GGMs from observations while mitigating both connectivity and correlation biases (see Section 2). Assume that we observe $n$ samples from the distribution $\mathcal{N}(\mathbf{0}, \boldsymbol{\Sigma}_0)$ collected in the data matrix $\mathbf{X} \in \mathbb{R}^{n \times p}$. To estimate fair and sparse precision matrices from data, we adapt the celebrated graphical lasso method [26, 31, 32], a penalized maximum likelihood approach for recovering GGMs.

Given the sample covariance matrix $\hat{\boldsymbol{\Sigma}} = \frac{1}{n} \mathbf{X}^\top \mathbf{X}$, we present *Fair GLASSO*, a version of graphical lasso for fair GGMs,

$$\boldsymbol{\Theta}^* = \underset{\boldsymbol{\Theta}}{\operatorname{argmin}} \ \operatorname{tr}(\hat{\boldsymbol{\Sigma}} \boldsymbol{\Theta}) - \log \det(\boldsymbol{\Theta} + \epsilon \mathbf{I}) + \mu_1 \|\boldsymbol{\Theta}_{\bar{\mathcal{D}}}\|_1 + \mu_2 R_H(\boldsymbol{\Theta})$$

$$\text{s.t.} \quad \boldsymbol{\Theta} \in \mathcal{M} := \{\boldsymbol{\Theta} \in \mathbb{R}^{p \times p} : \boldsymbol{\Theta} \succeq 0, \ \|\boldsymbol{\Theta}\|_2^2 \leq \alpha\}, \tag{4}$$

where $R_H$ denotes a bias penalty measuring the fairness of $\boldsymbol{\Theta}$ and $\mu_1, \mu_2 \geq 0$ tune the encouragement of sparse and fair precision matrices, respectively. For the penalty $R_H$, we can choose not only our

proposed metrics $H$ and $H_{\mathrm{node}}$ but also any metric for measuring bias on graphs. Similar to existing works on graph Laplacian GGMs [33], the addition of $\epsilon \mathbf{I}$ for $\epsilon > 0$ adds practicality to our approach, permitting us to obtain positive semi-definite precision matrices. The ability to estimate rank-deficient matrices allows for disconnected graph solutions. We assume that the true precision matrix $\boldsymbol{\Theta}_0$ has bounded eigenvalues (see AS2 and AS3 in Section 3.2), hence the constraint $\|\boldsymbol{\Theta}\|_2^2 \leq \alpha$ on the spectral norm of $\boldsymbol{\Theta}$ for large enough $\alpha > 0$. In practice, an effective $\alpha$ can be obtained by overshooting its value based on the minimum eigenvalue of the sample covariance $\hat{\boldsymbol{\Sigma}}$. For further context of how both our proposed bias metrics $H$ and $H_{\mathrm{node}}$ and our Fair GLASSO method relate to existing works, we provide a detailed review of related works in Appendix A. We present our approach for estimating GGMs, but indeed we may consider other distributions for the problem formulation in (4), such as the Ising negative log-likelihood. As our theoretical analysis requires Gaussianity, we proceed under this assumption, but future work will see the application of fair regularization to other distributions. Moreover, our empirical results in Section 4 show satisfactory performance optimizing (4) even for real-world datasets with non-Gaussian data.

## 3.2 Fair GLASSO Theoretical Analysis

We theoretically characterize the performance of Fair GLASSO. In particular, we focus on the effect of our fairness penalty in (4). Our result demonstrates the error rate of $\boldsymbol{\Theta}^*$ not only from a traditional statistical perspective but also in terms of the bias in the true precision matrix $\boldsymbol{\Theta}_0$. Indeed, as $\boldsymbol{\Theta}_0$ becomes more unfair, we expect that imposing unbiased estimates hinders estimation performance. Let the set $\mathcal{S} := \{(i,j) \in [p]^2 : [\boldsymbol{\Theta}_0]_{ij} \neq 0, \ i \neq j\}$ contain the indices of the non-zero, off-diagonal entries of $\boldsymbol{\Theta}_0$. We first share the following assumptions on $\boldsymbol{\Theta}_0$ and $\mathbf{Z}$.

**AS1** *(Bounded sparsity) There exists a constant $s > 0$ such that the cardinality of $\mathcal{S}$ satisfies $|\mathcal{S}| \leq s$.*

**AS2** *(Bounded spectrum) There exists a constant $\underline{k} > 0$ such that $\lambda_{\min}(\boldsymbol{\Sigma}_0) \geq \underline{k} > 0$.*

**AS3** *(Bounded spectrum) There exists a constant $\bar{k} > 0$ such that $\lambda_{\max}(\boldsymbol{\Sigma}_0) \leq \bar{k} < \infty$.*

**AS4** *(Persistent groups) All groups have the same size, that is, $p_a = \bar{p} \geq 2$ for every $a \in [g]$.*

Assumptions AS1, AS2, and AS3 follow those from the distinguished work [32]. Note that AS4 is imposed for simplicity, but similar results hold if we merely require asymptotically similar groups sizes, where no groups vanish as $p \to \infty$. With our assumptions in place, we present our main result on the error rate of Fair GLASSO, the proof of which can be found in Appendix B.

**Theorem 1** *Assume that AS1 to AS4 hold and that $\mu_1 \asymp \sqrt{(\log p)/n}$ and $\mu_2 = o(1)$. Moreover, let $R_H = H$ from (2) and $\epsilon = 0$ in (4). With probability tending to 1 as $n, p \to \infty$, there exist constants $m_1, m_2 > 0$ such that*

$$\|\boldsymbol{\Theta}^* - \boldsymbol{\Theta}_0\|_F \leq m_1 \sqrt{\frac{(p+s)\log p}{n}} + m_2 \frac{\sqrt{g}\sqrt[4]{H(\boldsymbol{\Theta}_0)}}{\sqrt{p}}. \tag{5}$$

*Moreover, there exists a constant $q > 0$ such that if $\mu_2$ satisfies*

$$\mu_2^2 \leq \frac{qp^2 \log p}{g^2 n \sqrt{H(\boldsymbol{\Theta}_0)}}, \tag{6}$$

*then with probability tending to 1 as $p \to \infty$ we can further guarantee that*

$$\|\boldsymbol{\Theta}^* - \boldsymbol{\Theta}_0\|_F \leq m_1 \sqrt{\frac{(p+s)\log p}{n}}. \tag{7}$$

Our error bound consists of the Frobenius norm convergence rate for graphical lasso in [32] and a term accounting for the bias penalty in (4). In particular, the second term in (5) portrays the influence of bias in the true precision matrix $\boldsymbol{\Theta}_0$. Theorem 1 not only provides an intuitive error bound for fair estimation of GGMs but also exemplifies when a tradeoff between fairness and accuracy may occur. When the true model $\boldsymbol{\Theta}_0$ is biased, that is, $H(\boldsymbol{\Theta}_0)$ is large, then performance may suffer according to (5). However, if bias mitigation is mild enough, that is, if $\mu_2$ is small enough to satisfy (6), then we instead enjoy the error rate of [32] with no adverse effect from the bias penalty. Indeed, as the true

$\boldsymbol{\Theta}_0$ becomes fairer, so too grows the range of values of $\mu_2$ that guarantee (7). Thus, if $\boldsymbol{\Theta}_0$ is unbiased, then imposing a strong bias penalty can obtain fair estimates $\boldsymbol{\Theta}^*$ while maintaining accuracy.

In addition to the explicit Frobenius error rate of $\boldsymbol{\Theta}^*$, we are also interested in when we can sufficiently describe the true model behavior. Our next result shows how well Fair GLASSO solutions approximate the true distribution, which enjoys the same rate as in Theorem 1, proven in Appendix C.

**Corollary 1** *Let the assumptions of Theorem 1 hold. Then, with probability tending to 1 as $n, p \to \infty$, there exist constants $m_1', m_2' > 0$ such that*

$$\|\boldsymbol{\Theta}^*\boldsymbol{\Sigma}_0 - \mathbf{I}\|_F \le m_1'\sqrt{\frac{(p+s)\log p}{n}} + m_2'\frac{\sqrt{g}\sqrt[4]{H(\boldsymbol{\Theta}_0)}}{\sqrt{p}}. \tag{8}$$

*Moreover, there exists a constant $q > 0$ such that when $\mu_2$ satisfies (6), then with probability tending to 1 as $p \to \infty$ we can further guarantee that*

$$\|\boldsymbol{\Theta}^*\boldsymbol{\Sigma}_0 - \mathbf{I}\|_F \le m_1'\sqrt{\frac{(p+s)\log p}{n}}. \tag{9}$$

Note that a similar rate to (9) holds up to a constant if we replace $\boldsymbol{\Sigma}_0$ with $\hat{\boldsymbol{\Sigma}}$. Thus, we may apply $\|\boldsymbol{\Theta}^*\hat{\boldsymbol{\Sigma}} - \mathbf{I}\|_F$ as an error metric when the true covariance matrix $\boldsymbol{\Sigma}_0$ is unavailable.

### 3.3 Algorithmic Implementation

If we choose $R_H$ in (4) as $H$ or $H_{\text{node}}$, the convexity of the resultant problem allows us to introduce a simple yet effective algorithm for Fair GLASSO estimates. We base our approach on an accelerated proximal gradient method known as fast iterative shrinkage algorithm (FISTA) [34], which is well suited to solving non-smooth, constrained optimization problems. More-

---

**Algorithm 1:** Fair GLASSO from Gaussian observations.

**Input:** Sample covariance $\hat{\boldsymbol{\Sigma}}$, weights $\mu_1$ and $\mu_2$, Lipschitz constant $L$ of $f$.

1 Initialize $\hat{\boldsymbol{\Theta}}^{(0)} = \check{\boldsymbol{\Theta}}^{(1)} \in \mathcal{M}$, $t^{(1)} = 1$, $k = 1$.
2 **while** *Stopping criteria not met* **do**
3 $\quad$ Proximal gradient descent: $\dot{\boldsymbol{\Theta}}^{(k)} = \mathcal{T}_{\mu_1/L}\left(\check{\boldsymbol{\Theta}}^{(k)} - \frac{1}{L}\nabla f(\check{\boldsymbol{\Theta}}^{(k)})\right)$.
4 $\quad$ Projection step: $\hat{\boldsymbol{\Theta}}^{(k)} = \Pi_{\mathcal{M}}(\dot{\boldsymbol{\Theta}}^{(k)})$.
5 $\quad$ Adaptive step size update: $t^{(k+1)} = \frac{1}{2}\left(1 + \sqrt{1 + 4(t^{(k)})^2}\right)$.
6 $\quad$ Accelerated update: $\check{\boldsymbol{\Theta}}^{(k+1)} = \hat{\boldsymbol{\Theta}}^{(k)} + \frac{t^{(k)}-1}{t^{(k+1)}}\left(\hat{\boldsymbol{\Theta}}^{(k)} - \hat{\boldsymbol{\Theta}}^{(k-1)}\right)$.
7 $\quad$ Update iteration: $k = k + 1$.
8 **end**

**Output :** Estimated precision matrix $\hat{\boldsymbol{\Theta}} = \hat{\boldsymbol{\Theta}}^{(k)}$.

---

over, our ensuing FISTA approach is still applicable under other distributions as long as the associated loss in (4) is convex and differentiable, such as the negative log-likelihood of the Ising model.

We separate the Fair GLASSO objective function $F(\boldsymbol{\Theta}) = f(\boldsymbol{\Theta}) + h(\boldsymbol{\Theta})$ into its smooth and non-smooth terms via $f(\boldsymbol{\Theta})$ and $h(\boldsymbol{\Theta})$, respectively, which are given by

$$f(\boldsymbol{\Theta}) := \text{tr}(\hat{\boldsymbol{\Sigma}}\boldsymbol{\Theta}) - \log\det(\boldsymbol{\Theta} + \epsilon\mathbf{I}) + \mu_2 R_H(\boldsymbol{\Theta}), \qquad h(\boldsymbol{\Theta}) := \mu_1\|\boldsymbol{\Theta}_{\bar{\mathcal{D}}}\|_1. \tag{10}$$

The proposed algorithm to estimate $\boldsymbol{\Theta}^*$ is presented in Algorithm 1. We discuss the steps of our algorithm in Appendix D, and we provide further details in Appendix E, including specifying the gradient $\nabla_{\boldsymbol{\Theta}} f$ and the Lipschitz constant of $f$ when $R_H = H$ or $R_H = H_{\text{node}}$.

Computationally, the complexity of Algorithm 1 is limited by an eigendecomposition in the projection step and a matrix inverse in the proximal gradient descent step (see (50) and (51) in Appendix D for details). Over-the-shelf implementations of these operations render a computational complexity of $\mathcal{O}(p^3)$. However, implementations based on fast matrix multiplication may result in an improved complexity of $\mathcal{O}(p^{2.4})$, a remarkable improvement since the optimization problem involves learning $p^2$ variables. Finally, in addition to a mild computational complexity, the proposed algorithm enjoys a convergence rate of $\mathcal{O}(\frac{1}{k^2})$, which we formally state next and prove in Appendix F.

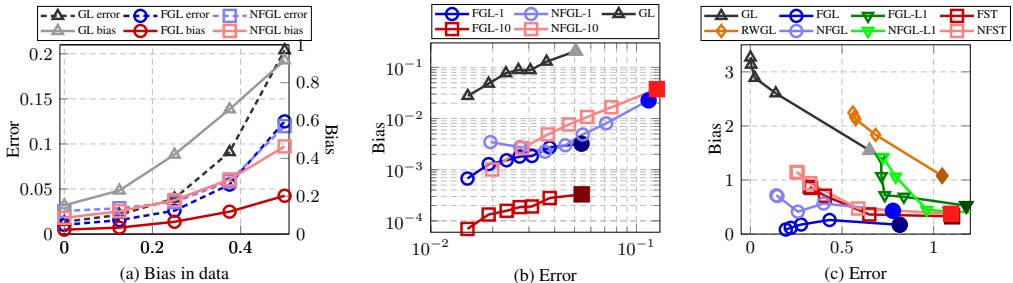

Figure 2: Estimation performance in terms of error and bias. (a) Bias and error for estimating a fair graph as data becomes more biased. (b) Bias and error as graph size $p$ grows for ER graphs. (c) Bias and error for a biased real-world network [23] as the number of observations $n$ grows.

**Theorem 2** *Let $\{\hat{\mathbf{\Theta}}^{(k)}\}_{k \geq 1}$ be the sequence generated by Algorithm 1 for solving the optimization problem* (4)*, where we denote the global minimum by $\mathbf{\Theta}^*$. Then, for any $k \geq 1$,*

$$\|\hat{\mathbf{\Theta}}^{(k)} - \mathbf{\Theta}^*\|_F^2 \leq \frac{4L\|\mathbf{\Theta}^{(0)} - \mathbf{\Theta}^*\|_F^2}{\alpha(k+1)^2}, \tag{11}$$

*where $L$ is the Lipschitz constant of $f(\mathbf{\Theta})$ and $\alpha$ corresponds to the spectral constraint in* (4)*.*

Thus, Theorem 2 guarantees convergence of Algorithm 1 to the optimal solution $\mathbf{\Theta}^*$ under our constraints in (4) with either of our bias metrics in (2) or (3). Not only are the convex fairness penalties amenable to efficient algorithms with well-understood performance guarantees, we also are able to guarantee that our algorithm converges with respect to the estimation variable, which is stronger than previous works' results on convergence of the objective function [35].

## 4   Experiments

We illustrate the ability of Fair GLASSO to reliably estimate both synthetic and real-world graphs from data while promoting unbiased connections. Extensive experimental details including our performance metrics, the baselines with which we compare, and the real-world datasets are provided in Appendix G; these details are summarized here. We include additional experiments on the effect of varying the hyperparameters $\mu_1$ and $\mu_2$ and violating assumptions (AS1)-(AS4) of Theorem 1 on Appendix H.

We compare our method with existing approaches for both scalability and performance. In particular, we consider (i) **GL**: Traditional graphical lasso [26], (ii) **FGL**: Fair GLASSO with $R_H = H$, (iii) **NFGL**: Fair GLASSO with $R_H = H_{\text{node}}$, (iv) **FST**: Network inference from spectral templates with a group-wise bias penalty [18, 36], (v) **NFST**: Network inference from spectral templates with a node-wise bias penalty [15, 18], and (vi) **RWGL**: Graphical lasso with randomly rewired edges.

We then perform GGM estimation on multiple real-world networks: (i) **Karate club**: the social network of Zachary's karate club members [23], (ii) **School**: A contact network of high school students [37], (iii) **Friendship**: The friendship network of the same high school students as in School [37], (iv) **Co-authorship**: An author collaboration network [38], and (v) **MovieLens**: A movie recommender network [39]. Figure 1 demonstrated that interconnected data may have fair or unfair relationships; thus, our experiment not only exemplifies the viability of our approach for real-world settings but also the fairness-accuracy tradeoff depending on biases in data.

### 4.1   Estimating Fair Graphs with Biased Data

Consider the realistic setting where our model is to be implemented in a fair setting, but our observations contain unfair biases [8]. We aim to obtain accurate graphical models by reducing the biases encoded in data. We consider synthetic networks whose nodes show no preferential connections, but our observations become increasingly unfair, growing in preference for within-group correlations.

Figure 2a presents the error and bias from networks estimated using graphical lasso with and without bias penalties $H$ and $H_{\text{node}}$. As expected, all methods show an increase in both error and bias as the

data becomes more unfair, as our observations are not only straying from the true distribution but also tending toward unfair behavior. However, **FGL** and **NFGL** not only preserve a lower bias than **GL**, but we also improve estimation performance. This significant result exemplifies the situation described in Section 3.2; our proposed penalties not only yield unbiased estimates but also serve as informative priors when the underlying graph is fair. Thus, we enjoy improvement in both fairness and accuracy for this realistic setting.

## 4.2 Performance as Graph Size Increases

Fair GLASSO adapts traditional GGM learning through the bias penalty, which includes (2) and (3). To observe the regularization effect of our penalties, we compare graphical lasso both with and without bias penalties for estimating synthetic networks as the graph size $p$ grows, which also demonstrates the scalability of our method. We thus implement **GL** via a state-of-the-art approach for comparison [26].

| Nodes $p$ | GL | FGL-0 | FGL-1 |
|---|---|---|---|
| 50 | 2.06 | 0.55 | 0.64 |
| 200 | 18.80 | 8.14 | 8.87 |
| 1000 | 10225.74 | 1900.89 | 1893.57 |

Table 1: Running time in seconds of Algorithm 1 and graphical lasso via [26].

Figure 2b shows the relationship between error and bias in the estimated graphs. Each line corresponds to a graph estimation method, and points on the lines denote the varying dimension, ranging from $p = 50$ (highlighted via darker, filled markers) to $p = 1000$.

First, observe that **GL** achieves superior accuracy at the expense of a larger bias, while **NFGL** improves bias, albeit with greater error. In contrast, **FGL** for $\mu_2 \in \{1, 10\}$ can improve bias without sacrificing accuracy, where $\mu_2 = 10$ yields the most Pareto-optimal solution. This result aligns with Theorem 1, showcasing the ability of Fair GLASSO to maintain estimation performance while significantly improving the fairness of the obtained graph. Critically, even as $p$ increases, our method enjoys relatively short running times, ranging from 0.5 seconds for 50 nodes to 30 minutes for 1000, which we show in Table 1. Our implementation of the classical algorithm in [26] requires 2 seconds and 170 minutes for $p = 50$ and $p = 1000$, respectively. We can then conclude that our efficient algorithm for Fair GLASSO can sufficiently handle larger graphs.

## 4.3 Social Network with Synthetic Signals

We next apply Fair GLASSO for the **Karate club** network, a real-world graph with known biased connections [23]. As this network famously exhibits group-wise modularity [22], we can compare different methods for estimating a real biased network. We show bias and error as the number of data samples increases from $n = 10^2$ (denoted by darker, filled markers) to $n = 10^5$ in Figure 2c. Since this graph does not have data, we generate synthetic Gaussian observations on the social network. Note that we only consider synthetic samples for this real-world dataset; the remainder are equipped with a set of real graph signals. In addition to the previously considered baselines, we also compare to **FGL-L1** and **NFGL-L1**, which correspond to Fair GLASSO using the group-wise and node-wise bias metrics in [18] as the penalty $R_H$. These metrics may prioritize balancing some group pairs over others, as described in Section 2.1.

For all methods, increasing the number of samples improves estimation error, but bias also grows since the underlying graph is unfair. Observe that all alternatives to **GL** are able to reduce estimation bias. As expected, randomly rewiring edges from graphical lasso estimates in **RWGL** does mildly improve bias when compared to **GL**, but error also rises significantly. The methods designed for fair graph estimation achieve the greatest improvement in bias, with our proposed methods **FGL** and **NFGL** outperforming **FST** and **NFST** in both bias and error. We also observe that **FGL** and **NFGL** using our bias metrics $H$ and $H_{\text{node}}$ with squared terms improve both fairness and accuracy over **FGL-L1** and **NFGL-L1** using the analogous metrics in [15, 18], which consider sums of absolute values of each term. Moreover, not only does **FGL** outperform other methods in both fairness and accuracy for all $n$, but **FGL** is the only approach that simultaneously decreases bias and error. Fair GLASSO is therefore viable for estimating real-world networks with known biased connections.

We also investigate the effect of $H$ versus $H_{\text{node}}$ for estimating group-wise modular networks. Figure 3 visualizes three graphs learned using **GL**, **FGL**, and **NFGL**. Both **FGL** and **NFGL** attempt to mitigate biased connections by reducing larger weights for existing within-group edges. However, since **FGL** aims to improve bias in expectation, Figure 3b shows an increase in negative within-group

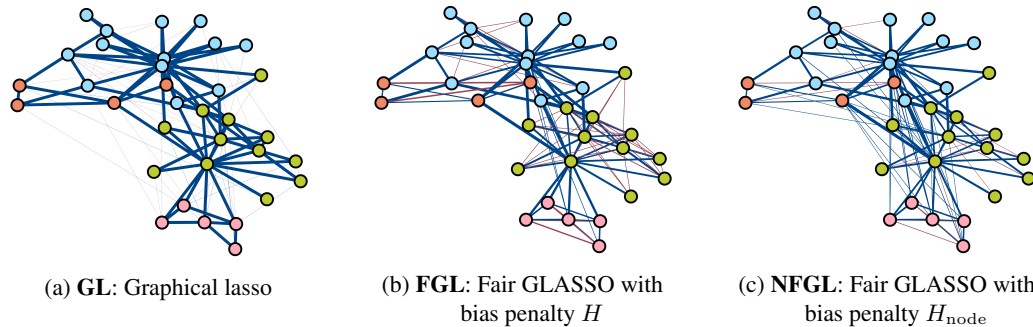

(a) **GL**: Graphical lasso

(b) **FGL**: Fair GLASSO with bias penalty $H$

(c) **NFGL**: Fair GLASSO with bias penalty $H_{\text{node}}$

Figure 3: Estimated **Karate club** network via graphical lasso with and without penalties $H$ and $H_{\text{node}}$. Node colors denote group membership, while edge thickness denotes edge weight magnitude and edge color its sign, with blue (red) as positive (negative) correlation. (a) Estimation via **GL**. (b) Estimation via **FGL**. (c) Estimation via **NFGL**.

edges, that is, negative partial correlations between nodes in the same group. Conversely, Figure 3c shows a network with more balanced connections *per node*, where more nodes are connected to new edges that are both positive and negative. This result suggests that the bias metric $H$ for group-wise balance in expectation aligns more with existing definitions of DP, while the node-wise DP gap $H_{\text{node}}$ behaves closer to an individual fairness metric [40, 41].

## 4.4 Fair GGMs for Real-World Data

Finally, in Table 2 we evaluate Fair GLASSO to estimate graphs from four real-world datasets consisting of two social networks, **School** and **Friendship** with *gender* as the sensitive attribute; a collaboration network **Co-authorship** with groups representing the *type of conference* in which each author publishes most; and a recommendation network **MovieLens**, where we consider binary sensitive attributes for each movie (node) denoting whether or not the movie was *released after 1991*. Evaluating Fair GLASSO on these datasets not only demonstrates its effectiveness on relevant real-world scenarios with biases, which are described in greater detail in Appendix G, but we can also showcase performance on non-Gaussian data, such as the discrete graph signals of the **School** and **Friendship** social networks. As each network varies in level of biases in their connections and observations, we show results for both weak and strong bias mitigation, that is, $\mu_2 \in \{1, 10^6\}$, for all fair graph learning methods.

For the relatively unbiased **School** and **Friendship** networks, our methods **FGL** and **NFGL** obtain superior estimation accuracy while sufficiently accounting for biases, particularly in comparison with **FST** and **NFST**. Unsurprisingly, we observe the lowest estimation error when $\mu_2 = 1$ is small enough such that **FGL** and **NFGL** achieve similar bias to **Ground truth**. However, observe that **FGL** and **NFGL** have low estimation error even for large $\mu_2 = 10^6$, which enjoys significant bias reduction. This verifies the results from Theorem 1 and in Figure 2a for real-world data; when the underlying graph is fair, our bias penalties serve as informative structural priors that improve performance.

For the **Co-authorship** network, we also observe the best accuracy using **FGL** and **NFGL** when $\mu_2$ is small enough that bias is similar to that of the true network. Critically, even when $\mu_2$ is large, we observe errors for **FGL** and **NFGL** competitive with **GL** while also achieving low bias. Moreover, for the **MovieLens** dataset, Fair GLASSO is the only method that rivals **GL** in accuracy while acquiring significantly fairer estimates. Indeed, **FGL** with $\mu_2 = 10^6$ is the only method to achieve both low error and bias simultaneously. This implies that the observations in both the **Co-authorship** and **MovieLens** datasets are biased, since high bias mitigation yielding fair estimates improves estimation performance. Thus, we demonstrate that relationships in real data can be explained by graphical models rivaling the accuracy of state-of-the-art approaches while also exhibiting fairer behavior.

| | School | | Co-authorship | | MovieLens | | Friendship | |
|---|---|---|---|---|---|---|---|---|
| | Error | Bias | Error | Bias | Error | Bias | Error | Bias |
| **Ground truth** | – | 0.2030 | – | 14.052 | – | 0.6791 | – | 0.1487 |
| **GL** | 0.2661 | 0.3111 | 0.1995 | 12.6102 | **0.0223** | 0.9529 | 0.6477 | 0.4068 |
| **RWGL-150** | 0.3497 | 0.3943 | 0.2308 | 12.5836 | 2.1919 | 0.9409 | 0.6509 | 0.4068 |
| **RWGL-300** | 0.3775 | 0.5110 | 0.2978 | 12.5735 | 2.2002 | 0.9184 | 0.6633 | 0.3861 |
| **FST** ($\mu_2 = 1$) | 0.4383 | 0.8942 | 0.4188 | 0.5754 | 0.1724 | 4.0568 | 1.0606 | 0.3787 |
| **NFST** ($\mu_2 = 1$) | 0.4386 | 0.8924 | 0.4068 | 6.6887 | 0.1724 | 4.0568 | 1.1149 | 0.3734 |
| **FST** ($\mu_2 = 10^6$) | 1.7820 | 1.0767 | 1.0801 | 129.5384 | 0.1724 | 4.0568 | 1.1924 | **0.0052** |
| **NFST** ($\mu_2 = 10^6$) | 1.6131 | 3.1971 | 1.0500 | 176.6285 | 0.1724 | 4.0568 | 1.1852 | 0.0081 |
| **FGL** ($\mu_2 = 1$) | **0.1417** | 0.4824 | **0.1896** | 10.3317 | 0.0253 | 0.8177 | **0.0505** | 0.1657 |
| **NFGL** ($\mu_2 = 1$) | **0.1417** | 0.4824 | **0.1895** | 11.8391 | 0.0253 | 0.8177 | **0.0505** | 0.1657 |
| **FGL** ($\mu_2 = 10^6$) | 0.1449 | **0.0308** | 0.2432 | **0.1899** | 0.0248 | **0.6106** | 0.0516 | 0.0153 |
| **NFGL** ($\mu_2 = 10^6$) | 0.2981 | 0.0827 | 0.2708 | 0.7908 | 0.0239 | 0.7104 | 0.0873 | 0.0243 |

Table 2: Bias and error for estimating four real-world networks. The top row shows the bias present in the true underlying network. The best performances are in **bold**.

## 5    Conclusion

This work proposes two metrics to evaluate bias in graphical models, which we apply as regularizers for fair GGM estimation. In particular, we adapt DP to measure biases in the conditional dependence structure encoded in graphical models, where nodes may show preferences for certain groups in terms of either connections or correlations. Unlike existing works that typically only consider fairness based on the unweighted topology of a known graph, we extend the concept of graph DP for the weighted connectivity patterns represented by the precision matrix of a Gaussian distribution. Moreover, we apply our group fairness for graphical models to modify graphical lasso for estimating fair GGMs. Future work will see more general graphical models, along with other extensions both in terms of the graph setting and fairness, which we discuss further in Appendix I.

## Broader Impact

In this work, we proposed a fair adaptation of graphical lasso, an extremely prominent method for complex data analysis. The development of methods that encourage fairness is necessary to ensure ethical and trustworthy tools, particularly those applied as extensively as GGMs to several critical and sensitive applications. Biases present in real-world graphs are well known, such as biased connections due to gender in social network analysis or segregated communities of co-authors in different disciplines. We revealed that these graphical biases extend beyond preferences in connections to include within-group correlations in behavior. Indeed, while intuitive, the tendency for group members to behave similarly has not been investigated for graphical models, as bias in signed edges has not been considered. Our paper contributes to expanding available unbiased graph-based methods, leading to extensions of other graphical models and statistical tools.

Moreover, as fairness on graphs is still nascent, several graph-based tasks have yet to be considered under the lens of fairness. Indeed, models are typically encouraged to be unbiased with respect to independent entities, but recent years have seen greater attention paid to the treatment of data equipped with graphical relationships. We not only participate in this movement, but we also extend fairness on graphs by considering weighted and signed edges for graphical models encoding conditional dependencies. This paper serves as a critical step in developing fair graph-based tools, particularly as GGMs are used in several high-stakes fields, including finance and medicine.

## Acknowledgments and Disclosure of Funding

This work was partially supported by the Spanish AEI PID2022-136887NB-I00 and TED2021-130347B-I00, the Community of Madrid via the ELLIS Madrid Unit, and the U.S. NSF under award CCF-2340481. Research was sponsored by the Army Research Office and was accomplished under Grant Number W911NF-17-S-0002. The views and conclusions contained in this document are those of the authors and should not be interpreted as representing the official policies, either expressed or implied, of the Army Research Office or the U.S. Army or the U.S. Government. The U.S. Government is authorized to reproduce and distribute reprints for Government purposes notwithstanding any copyright notation herein.

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

# A Related Work

## A.1 Graph Estimation

Obtaining graphical representations from data has long stood as one of the most prominent tasks in several fields, including statistics, GSP, and more [3, 42, 43]. Maximum likelihood estimation of GGMs was first introduced by Dempster [25], and subsequent additions of the $\ell_1$-norm penalty produced the famous graphical lasso approach [31, 32, 44]. Several modifications and extensions followed, typically proposing alternative penalties [4, 45–49], and its popularity has brought about copious theoretical investigation into its performance, its limitations, and more [26, 32, 50–52]. While Gaussianity is typical for estimating graphs, extensions to non-Gaussian distributions are also well studied [53–55]. Similar approaches have found use in GSP, both under additional constraints and in more general settings [33, 36, 56, 57].

Most works learn graphs from data solely to preserve some notion of fidelity that may potentially be aided by prior structural assumptions [58, 59]. However, estimating graphs while improving performance in both accuracy and fairness in connections remains novel [18]. Moreover, to the best of our knowledge, we are the first to estimate graphical models from data while explicitly encouraging fairness in its connections, particularly in terms of both connectivity and correlation bias [16].

## A.2 Fairness on Graphs

Fairness in graph-based tasks has received much attention recently, particularly in the field of machine learning. In addition to group fairness [13, 14], some works consider other definitions such as individual or structural fairness [40, 41, 60]. The most prevalent tasks for fairness on graphs are link prediction [12, 14, 19, 61, 62] and node representation [12, 13, 62–64], where nodes must experience equal treatment regardless of their sensitive attributes. Some works directly alter graph structure to promote unbiased connections, but, unlike our work, this requires a known graph [14, 19, 29, 30, 61, 65].

A number of works consider creating fair graphs, which share our goal of obtaining graph representations that possess unbiased connectivity patterns [16–18, 66]. This includes fair graph generative models [66], which aims to learn distributions of graph data, whereas we learn graph structure from nodal observations. We note [16–18] as the only other works of which we are aware that estimate graphs from data while considering fair outcomes. However, for [16, 17], the task is fundamentally different as they aim to cluster nodes fairly without explicitly imposing fairness on graph structure. While the task in [18] is the same as ours, we differ not only in how our samples are modeled but also in our definition of fairness, which is specific to graphical models encoding conditional dependence. Moreover, we provide further theoretical results, including guarantees of both error rates and algorithmic convergence.

# B Proof of Theorem 1

Our proof of Theorem 1 is inspired by that of [32]. While some steps in this work are analogous to those of our result in Theorem 1, the presence of the bias penalty yields subtle differences. Thus, we elaborate on all steps of the proof to demonstrate deviations from the original result along with providing a self-contained result for clarity. We first require the following lemma [67, Lemma A.3], which allows us to bound differences between entries of estimated and true Gaussian covariance matrix entries with high probability.

**Lemma 1 (Bickel and Levina [67])** *For iid Gaussian random vectors $\mathbf{Z}_i \sim \mathcal{N}(\mathbf{0}, \mathbf{S})$ for $i \in [n]$ such that $\lambda_{\max}(\mathbf{S}) \leq \bar{k} < \infty$, we have that*

$$\mathbb{P}\left[\left|\sum_{\ell=1}^{n} Z_{i\ell} Z_{j\ell} - S_{ij}\right| \geq n\epsilon\right] \leq c_1 \exp\{-c_2 n\epsilon^2\}, \tag{12}$$

*where $\epsilon > 0$ is bounded, and $c_1$, $c_2$, and $\epsilon$ are dependent only on $\bar{k}$.*

We proceed with the proof of (5) in the statement of Theorem 1. Consider a function $Q$ that measures the difference in the objective of (4) given $\boldsymbol{\Theta} \in \mathcal{M}$ and the true precision matrix $\boldsymbol{\Theta}_0$,

$$
\begin{aligned}
Q(\boldsymbol{\Theta}) \; := \; & \operatorname{tr}(\hat{\boldsymbol{\Sigma}}\boldsymbol{\Theta}) - \log \det \boldsymbol{\Theta} + \mu_1 \|\boldsymbol{\Theta}_{\bar{\mathcal{D}}}\|_1 + \mu_2 H(\boldsymbol{\Theta}) \\
& - \operatorname{tr}(\hat{\boldsymbol{\Sigma}}\boldsymbol{\Theta}_0) + \log \det \boldsymbol{\Theta}_0 - \mu_1 \|[\boldsymbol{\Theta}_0]_{\bar{\mathcal{D}}}\|_1 - \mu_2 H(\boldsymbol{\Theta}_0) \\
= \; & \operatorname{tr}((\hat{\boldsymbol{\Sigma}} - \boldsymbol{\Sigma}_0)(\boldsymbol{\Theta} - \boldsymbol{\Theta}_0)) + \operatorname{tr}(\boldsymbol{\Sigma}_0(\boldsymbol{\Theta} - \boldsymbol{\Theta}_0)) - (\log \det \boldsymbol{\Theta} - \log \det \boldsymbol{\Theta}_0) \\
& + \mu_1 \big( \|\boldsymbol{\Theta}_{\bar{\mathcal{D}}}\|_1 - \|[\boldsymbol{\Theta}_0]_{\bar{\mathcal{D}}}\|_1 \big) + \mu_2 \big( H(\boldsymbol{\Theta}) - H(\boldsymbol{\Theta}_0) \big).
\end{aligned} \tag{13}
$$

We may also represent the objective difference in terms of the disparity between $\boldsymbol{\Theta}$ and $\boldsymbol{\Theta}_0$, that is, $G(\boldsymbol{\Delta}) = Q(\boldsymbol{\Theta}_0 + \boldsymbol{\Delta})$ for $\boldsymbol{\Delta} = \boldsymbol{\Theta} - \boldsymbol{\Theta}_0$, where we have

$$
\begin{aligned}
G(\boldsymbol{\Delta}) \; = \; & \operatorname{tr}((\hat{\boldsymbol{\Sigma}} - \boldsymbol{\Sigma}_0)\boldsymbol{\Delta}) + \operatorname{tr}(\boldsymbol{\Sigma}_0\boldsymbol{\Delta}) - \big( \log \det(\boldsymbol{\Theta}_0 + \boldsymbol{\Delta}) - \log \det \boldsymbol{\Theta}_0 \big) \\
& + \mu_1 \big( \|(\boldsymbol{\Theta}_0 + \boldsymbol{\Delta})_{\bar{\mathcal{D}}}\|_1 - \|[\boldsymbol{\Theta}_0]_{\bar{\mathcal{D}}}\|_1 \big) + \mu_2 \big( H(\boldsymbol{\Theta}_0 + \boldsymbol{\Delta}) - H(\boldsymbol{\Theta}_0) \big).
\end{aligned} \tag{14}
$$

We may then use $Q$ and $G$ to compare $\boldsymbol{\Theta}^*$ and $\boldsymbol{\Theta}_0$ via their difference under the objective function of our proposed graphical lasso problem (4).

By the definition of $\boldsymbol{\Theta}^* \in \mathcal{M}$ as the minimizer of (4), we have that $Q(\boldsymbol{\Theta}^*) \le 0$ and thus $G(\boldsymbol{\Delta}^*) \le 0$ for $\boldsymbol{\Delta}^* = \boldsymbol{\Theta}^* - \boldsymbol{\Theta}_0$. If we can show that $G(\boldsymbol{\Delta}) > 0$ for every $\boldsymbol{\Delta} \in \mathcal{M}$ such that

$$
\|\boldsymbol{\Delta}\|_F > m_1 r_n + m_2 \frac{g \sqrt[4]{H(\boldsymbol{\Theta}_0)}}{\sqrt{p}} \tag{15}
$$

where

$$
r_n := \sqrt{\frac{(p+s)\log p}{n}} \tag{16}
$$

for some constants $m_1, m_2 > 0$, then we know that $\|\boldsymbol{\Delta}^*\|_F$ satisfies (5) in Theorem 1, as desired. To this end, we obtain a lower bound of $G(\boldsymbol{\Delta})$ and determine the conditions under which we can guarantee that $G(\boldsymbol{\Delta}) > 0$ for any $\boldsymbol{\Delta} \in \mathcal{M}$ such that (15) holds.

**Trace difference.** We begin by addressing the first term of (14). These steps are analogous to those of [32]. By the triangle inequality, we define nonnegative values $t_1$ and $t_2$ such that

$$
\left| \operatorname{tr}((\hat{\boldsymbol{\Sigma}} - \boldsymbol{\Sigma}_0)\boldsymbol{\Delta}) \right| \; \le \; \left| \sum_{i \ne j} (\hat{\Sigma}_{ij} - [\boldsymbol{\Sigma}_0]_{ij})\Delta_{ij} \right| + \left| \sum_{i=1}^{p} (\hat{\Sigma}_{ii} - [\boldsymbol{\Sigma}_0]_{ii})\Delta_{ii} \right| =: t_1 + t_2, \tag{17}
$$

for which we provide an upper bound. To bound $t_1$, note that by Lemma 1, we may choose any constant $c_1 > 0$ such that

$$
|\hat{\Sigma}_{ij} - [\boldsymbol{\Sigma}_0]_{ij}| \; = \; \left| \frac{1}{n} \sum_{k=1}^{n} X_{ki} X_{kj} - [\boldsymbol{\Sigma}_0]_{ij} \right| \ge c_1 \sqrt{\frac{\log p}{n}} \tag{18}
$$

with probability at most $d_1 p^{-d_2 c_1^2}$ for constants $d_1, d_2 > 0$. Then, by the union sum inequality,

$$
\begin{aligned}
\mathbb{P}\left[ \max_{i \ne j} |\hat{\Sigma}_{ij} - [\boldsymbol{\Sigma}_0]_{ij}| \ge c_1 \sqrt{\frac{\log p}{n}} \right] \; & = \; \mathbb{P}\left[ \bigcup_{i \ne j} |\hat{\Sigma}_{ij} - [\boldsymbol{\Sigma}_0]_{ij}| \ge c_1 \sqrt{\frac{\log p}{n}} \right] \\
& \le \; \sum_{i \ne j} \mathbb{P}\left[ |\hat{\Sigma}_{ij} - [\boldsymbol{\Sigma}_0]_{ij}| \ge c_1 \sqrt{\frac{\log p}{n}} \right] \\
& \le \; d_1(p^2 - p)p^{-d_2 c_1^2} \le d_1 p^{-d_3}, 
\end{aligned} \tag{19}
$$

where $c_1 > \sqrt{2/d_2}$ and $d_3 > d_2 c_1^2 - 2$. We then apply the Cauchy-Schwarz inequality for the following bound on $t_1$,

$$
t_1 \le \max_{i \ne j} \left| \hat{\Sigma}_{ij} - [\boldsymbol{\Sigma}_0]_{ij} \right| \cdot \|\boldsymbol{\Delta}_{\bar{\mathcal{D}}}\|_1 \le c_1 \sqrt{\frac{\log p}{n}} \|\boldsymbol{\Delta}_{\bar{\mathcal{D}}}\|_1, \tag{20}
$$

whose probability tends to 1 as $p \to \infty$ as in the right-hand side of (19).

For $t_2$, we again apply the Cauchy-Schwarz inequality and Lemma 1 to obtain

$$
\begin{aligned}
t_2 &\leq \left( \sum_{i=1}^{p} (\hat{\Sigma}_{ii} - [\boldsymbol{\Sigma}_0]_{ii})^2 \right)^{1/2} \cdot \|\boldsymbol{\Delta}_{\mathcal{D}}\|_F \\
&\leq \sqrt{p} \max_{i \in [p]} |\hat{\Sigma}_{ii} - [\boldsymbol{\Sigma}_0]_{ii}| \cdot \|\boldsymbol{\Delta}_{\mathcal{D}}\|_F \\
&\leq c_2 \sqrt{\frac{p \log p}{n}} \cdot \|\boldsymbol{\Delta}_{\mathcal{D}}\|_F \\
&\leq c_2 r_n \|\boldsymbol{\Delta}_{\mathcal{D}}\|_F
\end{aligned}
\tag{21}
$$

with probability again approaching 1 as $p \to \infty$.

We combine (20) and (21) to obtain a lower bound of the first term of (14),

$$
\mathrm{tr}((\hat{\boldsymbol{\Sigma}} - \boldsymbol{\Sigma}_0)\boldsymbol{\Delta}) \geq -\left| \mathrm{tr}((\hat{\boldsymbol{\Sigma}} - \boldsymbol{\Sigma}_0)\boldsymbol{\Delta}) \right| \geq -c_1 \sqrt{\frac{\log p}{n}} \|\boldsymbol{\Delta}_{\bar{\mathcal{D}}}\|_1 - c_2 r_n \|\boldsymbol{\Delta}_{\mathcal{D}}\|_F .
\tag{22}
$$

**Log-determinant difference.** We next consider the difference of log determinants in $G(\boldsymbol{\Delta})$ which differs from the proof of [32] as we consider an inequality in (15) rather than an equality. Consider the function $f(t) = \log \det(\boldsymbol{\Theta}_0 + t\boldsymbol{\Delta})$. The derivative and second derivative of $f(t)$ are

$$
f'(t) = \mathrm{tr}((\boldsymbol{\Theta}_0 + t\boldsymbol{\Delta})^{-1}\boldsymbol{\Delta}),
\tag{23}
$$
$$
f''(t) = -\mathrm{tr}(\boldsymbol{\Delta}(\boldsymbol{\Theta}_0 + t\boldsymbol{\Delta})^{-1}\boldsymbol{\Delta}(\boldsymbol{\Theta}_0 + t\boldsymbol{\Delta})^{-1}),
\tag{24}
$$

respectively. Then, the Maclaurin series expansion of $f(1)$ with the integral form of the remainder is

$$
f(1) - f(0) = f'(0) + \int_0^1 f''(v)(1-v)dv,
\tag{25}
$$

so by the symmetry of $\boldsymbol{\Theta}_0$ and $\boldsymbol{\Delta}$ we apply the Kronecker product for

$\log \det(\boldsymbol{\Theta}_0 + \boldsymbol{\Delta}) - \log \det \boldsymbol{\Theta}_0$

$$
\begin{aligned}
&= \mathrm{tr}(\boldsymbol{\Sigma}_0\boldsymbol{\Delta}) - \int_0^1 (1-v)\mathrm{tr}(\boldsymbol{\Delta}(\boldsymbol{\Theta}_0 + v\boldsymbol{\Delta})^{-1}\boldsymbol{\Delta}(\boldsymbol{\Theta}_0 + v\boldsymbol{\Delta})^{-1})dv \\
&= \mathrm{tr}(\boldsymbol{\Sigma}_0\boldsymbol{\Delta}) - \mathrm{vec}(\boldsymbol{\Delta})^\top \left[ \int_0^1 (1-v)(\boldsymbol{\Theta}_0 + v\boldsymbol{\Delta})^{-1} \otimes (\boldsymbol{\Theta}_0 + v\boldsymbol{\Delta})^{-1}dv \right] \mathrm{vec}(\boldsymbol{\Delta}). \quad (26)
\end{aligned}
$$

Recall that the definition of the smallest eigenvalue of a matrix $\mathbf{A}$ is $\lambda_{\min}(\mathbf{A}) = \min_{\mathbf{x}:\|\mathbf{x}\|_2=1} \mathbf{x}^\top \mathbf{A}\mathbf{x}$. We thus have

$\log \det(\boldsymbol{\Theta}_0 + \boldsymbol{\Delta}) - \log \det \boldsymbol{\Theta}_0$

$$
\begin{aligned}
&\leq \mathrm{tr}(\boldsymbol{\Sigma}_0\boldsymbol{\Delta}) - \|\boldsymbol{\Delta}\|_F^2 \lambda_{\min}\left( \int_0^1 (1-v)(\boldsymbol{\Theta}_0 + v\boldsymbol{\Delta})^{-1} \otimes (\boldsymbol{\Theta}_0 + v\boldsymbol{\Delta})^{-1}dv \right) \\
&\leq \mathrm{tr}(\boldsymbol{\Sigma}_0\boldsymbol{\Delta}) - \|\boldsymbol{\Delta}\|_F^2 \int_0^1 (1-v)\lambda_{\min}^2(\boldsymbol{\Theta}_0 + v\boldsymbol{\Delta})^{-1}dv
\end{aligned}
\tag{27}
$$

since the eigenvalues of the Kronecker product of two matrices are the products of their eigenvalues. Then, we have that

$$
\begin{aligned}
\log \det(\boldsymbol{\Theta}_0 + \boldsymbol{\Delta}) - \log \det \boldsymbol{\Theta}_0 &\leq \mathrm{tr}(\boldsymbol{\Sigma}_0\boldsymbol{\Delta}) - \frac{1}{2} \|\boldsymbol{\Delta}\|_F^2 \min_{v \in [0,1]} \lambda_{\min}^2(\boldsymbol{\Theta}_0 + v\boldsymbol{\Delta})^{-1} \\
&\leq \mathrm{tr}(\boldsymbol{\Sigma}_0\boldsymbol{\Delta}) - \frac{1}{2} \|\boldsymbol{\Delta}\|_F^2 \lambda_{\min}^2(\boldsymbol{\Theta}_0 + \boldsymbol{\Delta})^{-1} \\
&= \mathrm{tr}(\boldsymbol{\Sigma}_0\boldsymbol{\Delta}) - \frac{1}{2} \|\boldsymbol{\Delta}\|_F^2 \lambda_{\max}^{-2}(\boldsymbol{\Theta}_0 + \boldsymbol{\Delta}) \\
&\leq \mathrm{tr}(\boldsymbol{\Sigma}_0\boldsymbol{\Delta}) - \frac{1}{2} \|\boldsymbol{\Delta}\|_F^2 (\|\boldsymbol{\Theta}_0\|_2 + \|\boldsymbol{\Delta}\|_2)^{-2} \\
&\leq \mathrm{tr}(\boldsymbol{\Sigma}_0\boldsymbol{\Delta}) - \frac{1}{2} \|\boldsymbol{\Delta}\|_F^2 (\underline{k}^{-1} + \|\boldsymbol{\Delta}\|_F)^{-2}.
\end{aligned}
\tag{28}
$$

We define $\tau := \max\{4, (1 + \underline{k}\|\boldsymbol{\Delta}\|_F)^2\}$, which gives us

$$
\begin{aligned}
\log\det(\boldsymbol{\Theta}_0 + \boldsymbol{\Delta}) - \log\det\boldsymbol{\Theta}_0 \;&\leq\; \mathrm{tr}(\boldsymbol{\Sigma}_0\boldsymbol{\Delta}) - \frac{1}{2}\|\boldsymbol{\Delta}\|_F^2 \left(\underline{k}^{-1}\max\{2, \underline{k}\|\boldsymbol{\Delta}\|_F + 1\}\right)^{-2} \\
&\leq\; \mathrm{tr}(\boldsymbol{\Sigma}_0\boldsymbol{\Delta}) - \frac{1}{2\tau}\underline{k}^2\|\boldsymbol{\Delta}\|_F^2.
\end{aligned}
\tag{29}
$$

**Sparsity penalties.** For the sparsity penalties, note that by the definition of $\mathcal{S}$, we may follow the steps in [32] and exploit the fact that $\|[\boldsymbol{\Theta}_0 + \boldsymbol{\Delta}]_{\bar{\mathcal{D}}}\|_1 = \|[\boldsymbol{\Theta}_0 + \boldsymbol{\Delta}]_{\bar{\mathcal{D}}\cap\mathcal{S}}\|_1 + \|\boldsymbol{\Delta}_{\bar{\mathcal{D}}\cap\bar{\mathcal{S}}}\|_1$ and $\|[\boldsymbol{\Theta}_0]_{\bar{\mathcal{D}}}\|_1 = \|[\boldsymbol{\Theta}_0]_{\bar{\mathcal{D}}\cap\mathcal{S}}\|_1$. Then, by the triangle inequality,

$$
\mu_1(\|[\boldsymbol{\Theta}_0 + \boldsymbol{\Delta}]_{\bar{\mathcal{D}}}\|_1 - \|[\boldsymbol{\Theta}_0]_{\bar{\mathcal{D}}}\|_1) \geq \mu_1(\|\boldsymbol{\Delta}_{\bar{\mathcal{D}}\cap\bar{\mathcal{S}}}\|_1 - \|\boldsymbol{\Delta}_{\bar{\mathcal{D}}\cap\mathcal{S}}\|_1).
\tag{30}
$$

**DP gaps.** Finally, we consider the difference in DP gaps. Observe that $H(\boldsymbol{\Theta})$ is both differentiable and convex in $\boldsymbol{\Theta}$. Thus, we have that

$$
\begin{aligned}
H(\boldsymbol{\Theta}_0 + \boldsymbol{\Delta}) - H(\boldsymbol{\Theta}_0) \;&\geq\; \mathrm{tr}(\nabla_{\boldsymbol{\Theta}}H(\boldsymbol{\Theta}_0)\boldsymbol{\Delta}) \\
&\geq\; -\left|\mathrm{tr}(\nabla_{\boldsymbol{\Theta}}H(\boldsymbol{\Theta}_0)\boldsymbol{\Delta})\right| \\
&\geq\; -\left|\mathrm{tr}(\nabla_{\boldsymbol{\Theta}}H(\boldsymbol{\Theta}_0)\boldsymbol{\Theta}_0)\right| - \left|\mathrm{tr}(\nabla_{\boldsymbol{\Theta}}H(\boldsymbol{\Theta}_0)\boldsymbol{\Theta})\right| \\
&\geq\; -\|\nabla_{\boldsymbol{\Theta}}H(\boldsymbol{\Theta}_0)\|_* (\|\boldsymbol{\Theta}_0\|_2 + \|\boldsymbol{\Theta}\|_2) \\
&\geq\; -\left(\alpha^{1/2} + \underline{k}^{-1}\right)\|\nabla_{\boldsymbol{\Theta}}H(\boldsymbol{\Theta}_0)\|_*.
\end{aligned}
\tag{31}
$$

Moreover, recall that by definition $\mathbf{C}_{ab}$ is block-wise constant for every $a, b \in [g]$. Thus, the gradient $\nabla_{\boldsymbol{\Theta}}H(\boldsymbol{\Theta}_0)$ as in the right-hand side of (51) is block-wise constant with $g$ blocks, hence it is at most rank $g$. We then have that

$$
\begin{aligned}
\|\nabla_{\boldsymbol{\Theta}}H(\boldsymbol{\Theta}_0)\|_* \;&\leq\; \sqrt{g}\,\|\nabla_{\boldsymbol{\Theta}}H(\boldsymbol{\Theta}_0)\|_F \\
&\leq\; \frac{2\sqrt{g}}{g^2 - g}\sum_{a=1}^{g}\sum_{b\neq a}\mathrm{tr}(\mathbf{C}_{ab}\boldsymbol{\Theta}_0)\|\mathbf{C}_{ab}\|_F \\
&=\; \frac{2\sqrt{g}}{g^2 - g}\sum_{a=1}^{g}\sum_{b\neq a}|\mathrm{tr}(\mathbf{C}_{ab}\boldsymbol{\Theta}_0)|\left(\frac{1}{\bar{p}^2 - \bar{p}} + \frac{1}{\bar{p}^2}\right)^{1/2} \\
&\leq\; \frac{4\sqrt{g}}{\bar{p}(g^2 - g)}\sum_{a=1}^{g}\sum_{b\neq a}|\mathrm{tr}(\mathbf{C}_{ab}\boldsymbol{\Theta}_0)| \\
&\leq\; \frac{4\sqrt{g}}{\bar{p}\sqrt{g^2 - g}}\left(\sum_{a=1}^{g}\sum_{b\neq a}|\mathrm{tr}(\mathbf{C}_{ab}\boldsymbol{\Theta}_0)|^2\right)^{1/2},
\end{aligned}
\tag{32}
$$

where the last inequality holds by $\|\mathbf{x}\|_1 \leq \sqrt{m}\|\mathbf{x}\|_2$ for any vector $\mathbf{x} \in \mathbb{R}^m$. By the definition of $H$,

$$
\|\nabla_{\boldsymbol{\Theta}}H(\boldsymbol{\Theta}_0)\|_* \;\leq\; \frac{4\sqrt{g}}{\bar{p}}\sqrt{H(\boldsymbol{\Theta}_0)}.
\tag{33}
$$

We then have the following lower bound for the difference in DP gaps,

$$
H(\boldsymbol{\Theta}_0 + \boldsymbol{\Delta}) - H(\boldsymbol{\Theta}_0) \;\geq\; -\frac{4\sqrt{g}(\sqrt{\alpha} + \underline{k}^{-1})}{\bar{p}}\sqrt{H(\boldsymbol{\Theta}_0)} \;\geq\; -\frac{c_3 g}{\bar{p}}\sqrt{H(\boldsymbol{\Theta}_0)}.
\tag{34}
$$

We now combine the lower bounds for each term in (14), that is, (22), (29), (30), and (34). For $\epsilon_1 < 1$, we let

$$
\mu_1 = \frac{c_1}{\epsilon_1}\sqrt{\frac{\log p}{n}}.
\tag{35}
$$

Then, we have that

$$
\begin{aligned}
G(\boldsymbol{\Delta}) \;\geq\;& \frac{1}{2\tau}\underline{k}^2 \left\|\boldsymbol{\Delta}\right\|_F^2 - c_1\sqrt{\frac{\log p}{n}}\left\|\boldsymbol{\Delta}_{\bar{\mathcal{D}}}\right\|_1 - c_2 r_n \left\|\boldsymbol{\Delta}_{\mathcal{D}}\right\|_F \\
&+ \mu_1(\left\|\boldsymbol{\Delta}_{\bar{\mathcal{D}}\cap\bar{\mathcal{S}}}\right\|_1 - \left\|\boldsymbol{\Delta}_{\bar{\mathcal{D}}\cap\mathcal{S}}\right\|_1) - \frac{\mu_2 c_3 g}{\bar{p}}\sqrt{H(\boldsymbol{\Theta}_0)}
\end{aligned}
$$

$$
\geq \frac{1}{2\tau}\underline{k}^2\left\|\boldsymbol{\Delta}\right\|_F^2 - c_1\left(1+\frac{1}{\epsilon_1}\right)r_n\left\|\boldsymbol{\Delta}_{\bar{\mathcal{D}}}\right\|_F - c_2 r_n\left\|\boldsymbol{\Delta}_{\mathcal{D}}\right\|_F - \frac{\mu_2 c_3 g}{\bar{p}}\sqrt{H(\boldsymbol{\Theta}_0)}
$$

$$
= \left\|\boldsymbol{\Delta}_{\bar{\mathcal{D}}}\right\|_F\left[\frac{1}{4\tau}\underline{k}^2\left\|\boldsymbol{\Delta}_{\bar{\mathcal{D}}}\right\|_F - c_1\left(1+\frac{1}{\epsilon_1}\right)r_n\right] \tag{36}
$$

$$
+ \left\|\boldsymbol{\Delta}_{\mathcal{D}}\right\|_F\left[\frac{1}{4\tau}\underline{k}^2\left\|\boldsymbol{\Delta}_{\mathcal{D}}\right\|_F - c_2 r_n\right] \tag{37}
$$

$$
+ \left[\frac{1}{4\tau}\underline{k}^2\left\|\boldsymbol{\Delta}\right\|_F^2 - \frac{\mu_2 c_3 g}{\bar{p}}\sqrt{H(\boldsymbol{\Theta}_0)}\right], \tag{38}
$$

where we aim to find conditions on $\left\|\boldsymbol{\Delta}\right\|_F$ ensuring that each term (36), (37), and (38) are positive so that $G(\boldsymbol{\Delta}) > 0$. For the first two terms (36) and (37), we obtain the following lower bounds

$$
\left\|\boldsymbol{\Delta}_{\bar{\mathcal{D}}}\right\|_F > 4\tau\underline{k}^{-2}c_1\left(1+\frac{1}{\epsilon_1}\right)r_n \tag{39}
$$

and

$$
\left\|\boldsymbol{\Delta}_{\mathcal{D}}\right\|_F > 4\tau\underline{k}^{-2}c_2 r_n, \tag{40}
$$

respectively. For the third term (38), we have

$$
\left\|\boldsymbol{\Delta}\right\|_F^2 > 4\tau c_3\mu_2\cdot\frac{g}{\bar{p}}\sqrt{H(\boldsymbol{\Theta}_0)}. \tag{41}
$$

All three conditions (40), (39), and (41) guarantee that $G(\boldsymbol{\Delta})$ is positive. Combining the conditions, we obtain a sufficient condition to ensure that $G(\boldsymbol{\Delta}) > 0$,

$$
\left\|\boldsymbol{\Delta}\right\|_F > \max\left\{4\tau\underline{k}^{-2}\left(c_1\left(1+\frac{1}{\epsilon_1}\right)+c_2\right)r_n, \sqrt{4\tau c_3\mu_2}\cdot\sqrt{\frac{g}{\bar{p}}}\sqrt[4]{H(\boldsymbol{\Theta}_0)}\right\}. \tag{42}
$$

Thus, there exist constants $m_1, m_2 > 0$ such that with high probability as $n, p \to \infty$,

$$
\left\|\boldsymbol{\Delta}\right\|_F > m_1 r_n + m_2\frac{\sqrt{g}\sqrt[4]{H(\boldsymbol{\Theta}_0)}}{\sqrt{\bar{p}}}. \tag{43}
$$

When this holds for any $\boldsymbol{\Delta} \in \mathcal{M}$, then we have shown that $G(\boldsymbol{\Delta}) > 0$. Thus, since $G(\boldsymbol{\Delta}^*) \leq 0$ and $\boldsymbol{\Delta}^* \in \mathcal{M}$, we have that

$$
\left\|\boldsymbol{\Delta}^*\right\|_F \leq m_1 r_n + m_2\frac{\sqrt{g}\sqrt[4]{H(\boldsymbol{\Theta}_0)}}{\sqrt{\bar{p}}} \tag{44}
$$

with high probability. Recalling that $g\bar{p} = p$ by AS4, we obtain the inequality in (5), as desired. Moreover, note that if we select $\mu_2$ such that

$$
\begin{aligned}
\mu_2 \;\leq\;& \frac{4\tau\underline{k}^{-4}}{c_3}\left(c_1\left(1+\frac{1}{\epsilon_1}\right)+c_2\right)^2\left(\frac{\bar{p}^2\log p}{n\sqrt{H(\boldsymbol{\Theta}_0)}}\right) \\
\leq\;& \frac{4\tau\underline{k}^{-4}}{c_3}\left(c_1\left(1+\frac{1}{\epsilon_1}\right)+c_2\right)^2\left(\frac{\bar{p}(p+s)\log p}{ng\sqrt{H(\boldsymbol{\Theta}_0)}}\right),
\end{aligned} \tag{45}
$$

then (42) becomes equivalent to

$$
\left\|\boldsymbol{\Delta}\right\|_F > 4\tau\underline{k}^{-2}\left(c_1\left(1+\frac{1}{\epsilon_1}\right)+c_2\right)r_n, \tag{46}
$$

and we need only satisfy $\left\|\boldsymbol{\Delta}\right\|_F > m_1 r_n$ with probability tending to 1 as $n, p \to \infty$ for any $\boldsymbol{\Delta} \in \mathcal{M}$, which guarantees that $G(\boldsymbol{\Delta}) > 0$. Again, since $G(\boldsymbol{\Delta}^*) \leq 0$, we then have with high probability as $p \to \infty$ that

$$
\left\|\boldsymbol{\Delta}^*\right\|_F \leq m_1 r_n \tag{47}
$$

for small enough $\mu_2$ dependent on the dimension $p$, the number of samples $n$, the number of groups $g$, and the fairness of the true precision matrix $H(\boldsymbol{\Theta}_0)$, satisfying (7) as desired.

## C   Proof of Corollary 1

Recall that by definition, $\boldsymbol{\Theta}_0 \boldsymbol{\Sigma}_0 = \mathbf{I}$. Then,

$$\|\boldsymbol{\Theta}^* \boldsymbol{\Sigma}_0 - \mathbf{I}\|_F = \|(\boldsymbol{\Theta}^* - \boldsymbol{\Theta}_0)\boldsymbol{\Sigma}_0\|_F \leq \bar{k}\|\boldsymbol{\Theta}^* - \boldsymbol{\Theta}_0\|_F. \tag{48}$$

The result then follows from Theorem 1 for $m_1' = \bar{k}m_1$ and $m_2' = \bar{k}m_2$.

## D   Optimization Algorithm Details

We provide an overview on computing the update steps for Algorithm 1. For each iteration, we first take a gradient step for $f(\boldsymbol{\Theta})$, where the Lipschitz constant of $f$ plays the role of the step size, then we perform a proximal step over the non-smooth terms in $h(\boldsymbol{\Theta})$. The proximal step for $h(\boldsymbol{\Theta})$ is the soft-thresholding operation for the $\ell_1$ norm,

$$\mathcal{T}_\lambda(\Theta_{ij}) = \max\{|\Theta_{ij}| - \lambda, 0\}\text{sign}(\Theta_{ij}), \tag{49}$$

where $\text{sign}(x)$ returns the sign of $x$. After the proximal gradient step, we perform an orthogonal projection $\Pi_{\mathcal{M}}$ onto the feasible set, which in our case is given by

$$\Pi_{\mathcal{M}}(\boldsymbol{\Theta}) = \mathbf{V} \min\left\{\max\left\{\boldsymbol{\Lambda}, 0\right\}, \alpha^{1/2}\right\} \mathbf{V}^\top, \tag{50}$$

with $\mathbf{V}$ and $\boldsymbol{\Lambda}$ respectively denoting the eigenvectors and eigenvalues of $\boldsymbol{\Theta}$, and with some abuse of notation we let $\min\{\max\{\boldsymbol{\Lambda}, 0\}, \alpha^{1/2}\}$ denote an element-wise minimum and maximum operation on the entries of $\boldsymbol{\Lambda}$, which projects all the eigenvalues in the diagonal of $\boldsymbol{\Lambda}$ onto the interval $[0, \alpha^{1/2}]$.

Both the gradient $\nabla_{\boldsymbol{\Theta}} f$ and the Lipschitz constant $L$ of $f$ are contingent on the choice of bias penalty. The following two lemmas provide their computation for the cases where we apply $R_H = H$, our DP gap measurement (2), or $R_H = H_{\text{node}}$, its node-wise alternative (3). The proofs of both lemmas are deferred to Appendix E.

**Lemma 2** *Let the bias penalty $R_H$ in (4) be given by $H(\boldsymbol{\Theta})$ in (2). Then, the gradient of $f(\boldsymbol{\Theta})$ is given by*

$$\nabla_{\boldsymbol{\Theta}} f(\boldsymbol{\Theta}) = \hat{\boldsymbol{\Sigma}} - (\boldsymbol{\Theta} + \epsilon\mathbf{I})^{-1} + \frac{2\mu_2}{g^2 - g} \sum_{a=1}^{g} \sum_{b \neq a} \text{tr}(\mathbf{C}_{ab}\boldsymbol{\Theta})\mathbf{C}_{ab}^\top \tag{51}$$

*where $\mathbf{C}_{ab} := [\mathbf{z}_a\mathbf{z}_a^\top/(p_a^2 - p_a) - \mathbf{z}_a\mathbf{z}_b^\top/(p_ap_b)]_{\bar{\mathcal{D}}}$ for every $a, b \in [g]$ such that $a \neq b$. Moreover, for $\bar{\mathbf{C}} := \sum_{a \neq b} \mathbf{C}_{ab} \otimes \mathbf{C}_{ab}^\top$ we have that $\nabla_{\boldsymbol{\Theta}} f(\boldsymbol{\Theta})$ is Lipschitz with constant*

$$L_1 = \frac{1}{\epsilon^2} + \frac{2\mu_2}{g^2 - g}\lambda_{\max}(\bar{\mathbf{C}}) \leq \bar{L}_1 = \frac{1}{\epsilon^2} + \frac{2\mu_2}{g^2 - g}\sum_{a=1}^{g}\sum_{b \neq a}\lambda_{\max}^2(\mathbf{C}_{ab}). \tag{52}$$

Intuitively, while $L_1$ is a smaller Lipschitz constant, it involves computing the eigendecomposition of a $p^2 \times p^2$ matrix, which may be prohibitive in higher-dimensional settings. Consequently, $\bar{L}_1$ provides an approximation involving only $p \times p$ matrices. The following lemma provides similar results when the node-wise DP gap $H_{\text{node}}$ is employed.

**Lemma 3** *Let the bias penalty $R_H$ in (4) be given by $H_{\text{node}}(\boldsymbol{\Theta})$ in (3). Then, the gradient of $f(\boldsymbol{\Theta})$ is given by*

$$\nabla_{\boldsymbol{\Theta}} f(\boldsymbol{\Theta}) = \hat{\boldsymbol{\Sigma}} - (\boldsymbol{\Theta} + \epsilon\mathbf{I})^{-1} + 2\mu_2[\mathbf{A}\boldsymbol{\Theta}_{\bar{\mathcal{D}}}]_{\bar{\mathcal{D}}}, \tag{53}$$

*where*

$$\mathbf{A} := \frac{1}{pg(g-1)^2}\sum_{a=1}^{g}\sum_{b \neq a}\left(\sum_{b \neq a}\frac{\mathbf{z}_a}{p_a} - \frac{\mathbf{z}_b}{p_b}\right)\left(\sum_{b \neq a}\frac{\mathbf{z}_a}{p_a} - \frac{\mathbf{z}_b}{p_b}\right)^\top. \tag{54}$$

*Moreover, $\nabla_{\boldsymbol{\Theta}} f(\boldsymbol{\Theta})$ is Lipschitz with constant*

$$L_2 = \frac{1}{\epsilon^2} + 2\mu_2\lambda_{\max}(\mathbf{A}). \tag{55}$$

For more concrete intuition, we present a brief analysis for bias metric $R_H(\boldsymbol{\Theta}) = H_{\text{node}}(\boldsymbol{\Theta})$, but a similar analysis holds for other penalties such as $R_H(\boldsymbol{\Theta}) = H(\boldsymbol{\Theta})$. The first step of Algorithm 1 is a proximal gradient step. Computing the gradient requires an inverse $(\boldsymbol{\Theta} + \epsilon \mathbf{I})^{-1}$ and product $\mathbf{A}\boldsymbol{\Theta}_{\bar{\mathcal{D}}}$, both incurring $\mathcal{O}(p^3)$ operations. The gradient step and soft-thresholding enjoy entry-wise computations with complexities $\mathcal{O}(p^2)$. The projection step onto the set of positive semidefinite matrices involves an eigendecomposition of $\dot{\boldsymbol{\Theta}}^{(k)}$ with complexity $\mathcal{O}(p^3)$. Finally, the step size update only requires scalar operations, and the accelerated update of $\dot{\boldsymbol{\Theta}}^{(k+1)}$ involves $\mathcal{O}(p^2)$ operations, so they can be neglected.

# E Gradients and Lipschitz Constants

Here, we provide the computation of relevant gradients and Lipschitz constants for $f$ in (10) used in the proposed Algorithm 1. Both the gradient and the Lipschitz constant of $f$ depend on the choice of bias penalty $H(\boldsymbol{\Theta})$.

## E.1 Proof of Lemma 2

First, we rewrite the demographic parity $\Delta\text{DP}$ in (2) as

$$H(\boldsymbol{\Theta}) = \frac{1}{g^2 - g} \sum_{a=1}^{g} \sum_{b \neq a} \text{tr}(\mathbf{C}_{ab}\boldsymbol{\Theta})^2, \tag{56}$$

where $\mathbf{C}_{ab}$ is defined in the statement of Lemma 2. The gradient of $H$ can be obtained as

$$\nabla_{\boldsymbol{\Theta}} H(\boldsymbol{\Theta}) = \frac{2}{g^2 - g} \sum_{a=1}^{g} \sum_{b \neq a} \text{tr}(\mathbf{C}_{ab}\boldsymbol{\Theta})\mathbf{C}_{ab}^{\top}, \tag{57}$$

and adding it to the gradient of the remaining terms of $f(\boldsymbol{\Theta})$, the result follows

$$\nabla_{\boldsymbol{\Theta}} f(\boldsymbol{\Theta}) = \hat{\boldsymbol{\Sigma}} - (\boldsymbol{\Theta} + \epsilon \mathbf{I})^{-1} + \frac{2\mu_2}{g^2 - g} \sum_{a=1}^{g} \sum_{b \neq a} \text{tr}(\boldsymbol{\Theta}\mathbf{C}_{ab})[\mathbf{C}_{ab}]^{\top}. \tag{58}$$

Next, to show that the gradient of $f(\boldsymbol{\Theta})$ is Lipschitz, it suffices to show that its Hessian is bounded. The Hessian $\nabla_{\boldsymbol{\Theta}}^2 f(\boldsymbol{\Theta})$ can be computed as

$$\nabla_{\boldsymbol{\Theta}}^2 f(\boldsymbol{\Theta}) = (\boldsymbol{\Theta} + \epsilon \mathbf{I})^{-1} \otimes (\boldsymbol{\Theta} + \epsilon \mathbf{I})^{-1} + \frac{2\mu_2}{g^2 - g} \sum_{a=1}^{g} \sum_{b \neq a} \left( \mathbf{C}_{ab} \otimes \mathbf{C}_{ab}^{\top} \right), \tag{59}$$

and it is bounded by

$$\|\nabla_{\boldsymbol{\Theta}}^2 f(\boldsymbol{\Theta})\|_2 \leq \left\| (\boldsymbol{\Theta} + \epsilon \mathbf{I})^{-1} \otimes (\boldsymbol{\Theta} + \epsilon \mathbf{I})^{-1} \right\|_2 + \frac{2\mu_2}{g^2 - g} \left\| \sum_{a=1}^{g} \sum_{b \neq a} \left( \mathbf{C}_{ab} \otimes \mathbf{C}_{ab}^{\top} \right) \right\|_2$$

$$\leq \frac{1}{\epsilon^2} + \frac{2\mu_2}{g^2 - g} \sum_{a=1}^{g} \sum_{b \neq a} \lambda_{\max}(\mathbf{C}_{ab}) = L_1. \tag{60}$$

Moreover, it also follows that

$$\|\nabla_{\boldsymbol{\Theta}}^2 f(\boldsymbol{\Theta})\|_2 \leq \left\| (\boldsymbol{\Theta} + \epsilon \mathbf{I})^{-1} \otimes (\boldsymbol{\Theta} + \epsilon \mathbf{I})^{-1} \right\|_2 + \frac{2\mu_2}{g^2 - g} \left\| \bar{\mathbf{C}} \right\|_2$$

$$\leq \frac{1}{\epsilon^2} + \frac{2\mu_2}{g^2 - g} \lambda_{\max}(\bar{\mathbf{C}}) = \tilde{L}_1. \tag{61}$$

Consequently, the gradient $\nabla_{\boldsymbol{\Theta}} f(\boldsymbol{\Theta})$ is Lipschitz with constants $L_1 \leq \tilde{L}_1$.

## E.2 Proof of Lemma 3

For the node-based DP gap $H_{\text{node}}(\boldsymbol{\Theta})$ as in (3), applying standard gradient calculus to compute the gradient and the Hessian of $f(\boldsymbol{\Theta})$ gives

$$\nabla_{\boldsymbol{\Theta}} f(\boldsymbol{\Theta}) = \hat{\boldsymbol{\Sigma}} - (\boldsymbol{\Theta} + \epsilon \mathbf{I})^{-1} + 2\mu_2 [\mathbf{A}\boldsymbol{\Theta}_{\bar{\mathcal{D}}}]_{\bar{\mathcal{D}}}, \tag{62}$$

$$\nabla_{\boldsymbol{\Theta}}^2 f(\boldsymbol{\Theta}) = (\boldsymbol{\Theta} + \epsilon \mathbf{I})^{-1} \otimes (\boldsymbol{\Theta} + \epsilon \mathbf{I})^{-1} + 2\mu_2 (\mathbf{I} \otimes \mathbf{A}). \tag{63}$$

Then, the Lipschitz constant is obtained from the following upper bound of the Hessian

$$\|\nabla_{\boldsymbol{\Theta}}^2 f(\boldsymbol{\Theta})\|_2 \leq (\boldsymbol{\Theta} + \epsilon \mathbf{I})^{-1} \otimes (\boldsymbol{\Theta} + \epsilon \mathbf{I})^{-1}\|_2 + 2\mu_2 \|\mathbf{I} \otimes \mathbf{A}\|_2 \leq \frac{1}{\epsilon^2} + 2\mu_2 \lambda_{\max}(A) = L_2. \tag{64}$$

# F  Proof of Theorem 2

Our proof builds over the convergence result for FISTA [34, Thm. 4.4], which establishes that the sequence $\{\boldsymbol{\Theta}^{(k)}\}_{k \geq 1}$ generated by FISTA satisfies the following bound

$$F(\boldsymbol{\Theta}^{(k)}) - F(\boldsymbol{\Theta}^*) \leq \frac{2L\|\boldsymbol{\Theta}^{(0)} - \boldsymbol{\Theta}^*\|_F^2}{(k+1)^2}, \tag{65}$$

where $\boldsymbol{\Theta}^*$ is the global minimum of the objective function of (4). This result requires $f(\boldsymbol{\Theta})$, the smooth components of $F(\boldsymbol{\Theta})$, to be a convex function with Lipschitz continuous gradient, while $h(\boldsymbol{\Theta})$, the non-smooth components of $F(\boldsymbol{\Theta})$, are only required to be convex.

From the expression of $f(\boldsymbol{\Theta})$ and $h(\boldsymbol{\Theta})$ in (10) it is clear that both terms of our objective function are convex. Furthermore, from Lemma 2 and Lemma 3 it follows that the gradient of $f(\boldsymbol{\Theta})$ is Lipschitz continuous when we consider $R_H = H$ or $R_H = H_{\text{node}}$, so the conditions from [34, Thm. 4.4] are met and (65) holds.

Next, we demonstrate that the objective function $F(\boldsymbol{\Theta})$ is strongly convex. A function $F(\boldsymbol{\Theta})$ is $\sigma$-strongly convex if $F(\boldsymbol{\Theta}) - \sigma\|\boldsymbol{\Theta}\|_F^2$ is also convex, which implies that $\nabla_{\boldsymbol{\Theta}}^2 F(\boldsymbol{\Theta}) - \sigma \mathbf{I} \succeq \mathbf{0}$. Put in words, the eigenvalues of the Hessian need to be larger than $\sigma$ for $F(\boldsymbol{\Theta})$ to be $\sigma$-strongly convex. Let $f_1(\boldsymbol{\Theta}) = \text{tr}(\hat{\boldsymbol{\Sigma}}\boldsymbol{\Theta}) - \log \det(\boldsymbol{\Theta} + \epsilon \mathbf{I})$, whose Hessian is given by

$$\nabla_{\boldsymbol{\Theta}}^2 f_1(\boldsymbol{\Theta}) = (\boldsymbol{\Theta} + \epsilon \mathbf{I})^{-1} \otimes (\boldsymbol{\Theta} + \epsilon \mathbf{I})^{-1}. \tag{66}$$

Since $\boldsymbol{\Theta}$ is constrained by $\|\boldsymbol{\Theta}\|_2^2 \leq \alpha$, applying the properties of the inverse and the Kronecker product renders the following bound on the eigenvalues of $\nabla_{\boldsymbol{\Theta}} f_1(\boldsymbol{\Theta})$

$$\lambda_{\min}\left(\nabla_{\boldsymbol{\Theta}}^2 f_1(\boldsymbol{\Theta})\right) = \frac{1}{\lambda_{\max}^2(\boldsymbol{\Theta} + \epsilon \mathbf{I})} \approx \frac{1}{\lambda_{\max}^2(\boldsymbol{\Theta})} = \frac{1}{\alpha}. \tag{67}$$

Recall that $\epsilon$ is assumed to be a small parameter such that $\epsilon^2 \ll \lambda_{\max}^2(\boldsymbol{\Theta})$. Consequently, $\nabla_{\boldsymbol{\Theta}}^2 f_1(\boldsymbol{\Theta}) \succeq \frac{1}{\alpha}\mathbf{I}$, so $f_1(\boldsymbol{\Theta})$ is strongly convex with constant $\frac{1}{\alpha}$, hence rendering $F(\boldsymbol{\Theta})$ also strongly convex with the same constant.

The last ingredient is given by [68, Thm. 5.25], which establishes that if $F(\boldsymbol{\Theta})$ is strongly convex with constant $\sigma$, then

$$\|\boldsymbol{\Theta} - \boldsymbol{\Theta}^*\|_F^2 \leq \frac{2}{\sigma}\left(F(\boldsymbol{\Theta}) - F(\boldsymbol{\Theta}^*)\right), \tag{68}$$

for every $\boldsymbol{\Theta}$ in the domain of $F$.

Finally, our result follows from combining (65) and (68).

# G  Experimental Details

The following details include descriptions of our datasets, baselines, and performance metrics. For synthetic experiments, we run simulations over 50 independent realizations of generated data. Moreover, excepting experiments for which we test different specific parameter values, we choose optimal values of hyperparameters via grid search.

**Datasets.** The numerical evaluation of the proposed algorithm is carried out over synthetic and real-world data. The main features of the different datasets are summarized in Table 3. Additional details are given next.

|  | Nodes (No.) | Edges | Signals (No.) | Groups | Sensitive attribute |
|---|---|---|---|---|---|
| **School** | Students (126) | 959 | Interactions (28561) | 2 | Gender |
| **Co-authorship** | Authors (130) | 525 | Keywords (1903) | 6 | Publication type |
| **MovieLens** | Movies (200) | 665 | Ratings (943) | 2 | Old/New |
| **Friendship** | Students (311) | 1009 | Interactions (47127) | 2 | Gender |

Table 3: Properties of the real datasets used in Section 4.

- **Synthetic data.** Unless specified otherwise, graphs with $p = 100$ nodes are sampled from an Erdős-Rényi (ER) random graph model with an average of 10 links per node. The precision matrix is set to either the combinatorial graph Laplacian [33] or an adjacency matrix with a loaded diagonal to ensure positive definiteness, and is employed to sample graph signals from a zero-mean multivariate Gaussian distribution. The number of sampled signals satisfies the conditions of Theorem 1. For experiments based on synthetic data, 50 independent realizations of graphs and signals are generated and we report the mean performance.

- **Karate club.** A social network where the 34 nodes represent members from a karate club and edges capture interactions between pairs of members outside the club [22, 23]. This dataset is famously modular with respect to groups, serving as a well-known biased network of real-world individuals. Since this graph does not have associated nodal observations, we generate signals as multivariate Gaussian samples as described in the previous point for synthetic data using the adjacency matrix of the social network with a loaded diagonal as the precision matrix. Similarly, we evaluate experiments with this synthetic data over 50 independent realizations of generated signals with the mean performance reported.

- **MovieLens**[1]**.** This movie-recommendation dataset contains ratings for 1,682 movies by 943 users, resulting in sparse data as many movies have few ratings. Biases in recommendation systems can reproduce and even exacerbate existing harmful stereotypes [69]. The MovieLens dataset, a common benchmark for fair graph machine learning, exemplifies our ability to form unbiased models from networks used for recommendation systems. To address this sparsity, we followed the setup in [17] and selected the 200 most-rated movies as nodes, using the ratings from the 943 users as graph signals. Since the dataset lacks a ground truth graph, we report the model fit as defined in Corollary 1 rather than the error in Table 2. The sensitive attribute for each node is determined by whether the movie was released before or after 1991.

- **Co-authorship**[2]**.**The dataset includes papers from the ACM conference, featuring 17,431 authors, 122,499 papers, and 1,903 keywords. The nodes represent a subset of these authors. The sensitive attribute associated with each author is determined by the conference type in which that author publishes the most. We demonstrate an example of data with more than two groups through the **Co-authorship** network with publication type as the sensitive attribute. To create the ground truth graph, we analyzed author-paper relationships and established an edge between two authors if they collaborated on a paper. For graph data generation, we utilized the total number of different keywords, with each input graph signal reflecting the frequency with which a specific author uses a particular keyword across their papers. We constructed the graph by selecting a connected subset of the authors. The associated graph signals were then used to estimate the graph through the considered methods.

- **School**[3]**.** The dataset contains the temporal network of contacts between students in a high school in Marseilles, France. In real-world social network analysis, common network characteristics such as homophily can lead to negative outcomes across gender in both social and academic settings [70]. Hence, both the **School** network and the **Friendship** network described below require scrutiny with respect to fairness, where gender is a critical consideration. The data includes the interactions of students from three classes over four days in December 2011. We used the available contact data to construct a ground truth graph, where nodes represent students and edges represent all interactions between them. We

---

[1]https://grouplens.org/

[2]https://dl.acm.org/

[3]http://www.sociopatterns.org/datasets/high-school-dynamic-contact-networks/

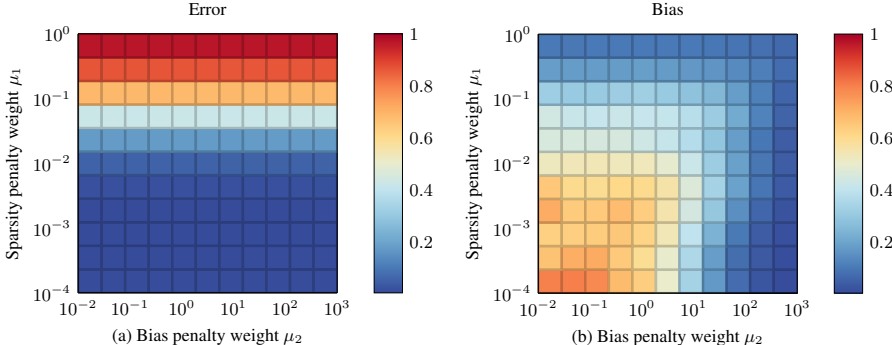

Figure 4: Performance in terms of error and bias for estimating a fair precision matrix. (a) Error as parameters $\mu_1$ and $\mu_2$ vary. (b) Bias as parameters $\mu_1$ and $\mu_2$ vary.

generated the signals by grouping the interactions into sets of four. Gender was considered as sensitive attribute for each node.

- **Friendship**[4]**.** This dataset corresponds to the contacts and friendship relations between students in nine classes at a high school in Marseilles, France, over five days in December 2013. Following the same procedure as with the **School** dataset, we used the available contact data to generate the ground truth graph, where nodes represent students and edges represent all interactions between them. The graph signals were generated by grouping the interactions into sets of four, and gender was considered the sensitive attribute.

**Baselines.** We compare the performance of our proposed Fair GLASSO approach with the following baselines:

- **GL**: The celebrated graphical lasso algorithm from [26] constitutes a workhorse alternative to estimate the topology of the graph encoded in precision matrices. However, it ignores the sensitive attributes of the nodes, so it is prone to include biases existing in the true graph.
- **RWGL**: a naive fair alternative to graphical lasso where several edges are randomly rewired after estimating the precision matrix with GL. Since the rewiring process is independent of the sensitive attributes of nodes, it will render a fairer estimate, but the random perturbation may yield highly inaccurate estimates. Methods denoted "**RWGL-N**" for some positive integer $N$ represents **RWGL** with $N$ edges rewired.
- **FST**: a fair alternative for learning the graph from stationary observations while mitigating biases in the topology via a group-wise bias penalty [18, 36]. Different from graphical lasso, stationary methods assume that the covariance of the observed data is a polynomial of a matrix encoding the graph topology. This more lenient assumption typically comes at the expense of requiring a larger number of observations.
- **NFST**: a variant of FST that estimates the graph topology from stationary observations including a node-wise regularization to promote fairness [15, 18].

Moreover, note that the notation "**FGL-X**" for some number **X** denotes **FGL** with $\mu_2 = \mathbf{X}$, and this similarly holds for "**NFGL-X**".

**Performance metrics.** To measure the error of our estimated precision matrices $\mathbf{\Theta}^*$, we apply the following normalized squared Frobenius error

$$\left\| \frac{[\mathbf{\Theta}^*]_{\bar{\mathcal{D}}}}{\|[\mathbf{\Theta}^*]_{\bar{\mathcal{D}}}\|_F} - \frac{[\mathbf{\Theta}_0]_{\bar{\mathcal{D}}}}{\|[\mathbf{\Theta}_0]_{\bar{\mathcal{D}}}\|_F} \right\|_F^2 \tag{69}$$

for true precision matrix $\mathbf{\Theta}_0$. When such a true matrix $\mathbf{\Theta}_0$ is unavailable, we instead apply the left-hand side of (8) from Corollary 1 using the sample covariance matrix $\hat{\mathbf{\Sigma}}$ estimated from observations.

To measure the bias in a given precision matrix $\mathbf{\Theta}^*$ for nodal group memberships $\mathbf{Z}$, observe that $H$ and $H_{\text{node}}$ compute average differences in edge weights within and across groups, whether

[4]http://www.sociopatterns.org/datasets/high-school-contact-and-friendship-networks/

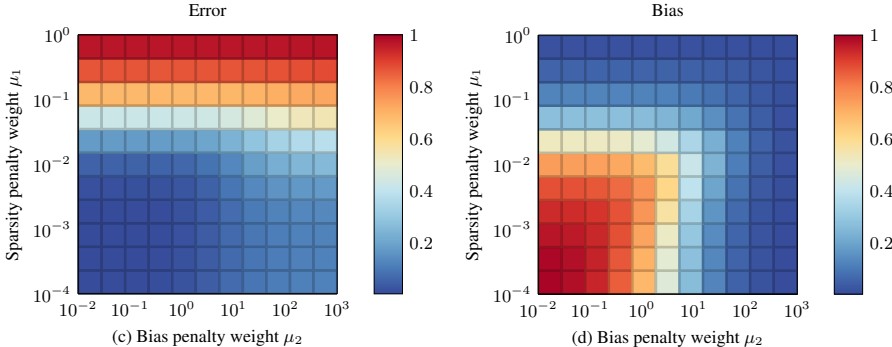

Figure 5: Performance in terms of error and bias for estimating an unfair precision matrix. (a) Error as parameters $\mu_1$ and $\mu_2$ vary. (b) Bias as parameters $\mu_1$ and $\mu_2$ vary.

group-wise or node-wise. Thus, for our practical bias metric we normalize by edge weight to obtain

$$\frac{2\sqrt{H(\mathbf{\Theta}^*)}}{\|[\mathbf{\Theta}^*]_{\bar{\mathcal{D}}}\|_1}, \tag{70}$$

that is, we take the square root of $H(\mathbf{\Theta})$, which acts as a squared $\ell_2$ norm across distinct group pairs, and we divide by the average edge weight for unordered variable pairs.

The group-wise modularity presented in Figure 1 is defined in [19], that is,

$$Q(\mathbf{\Theta}) = \sum_{a=1}^{g} \frac{\mathbf{z}_a^\top \mathbf{\Theta}_{\bar{\mathcal{D}}} \mathbf{z}_a}{2s} - \sum_{a=1}^{g} \left( \frac{\mathbf{z}_a^\top \mathbf{\Theta}_{\bar{\mathcal{D}}} \mathbf{z}_a}{2s} \right)^2, \tag{71}$$

where $s$ denotes the number of nonzero entries in $\mathbf{\Theta}$. To estimate partial correlation in Figure 1, we apply graphical lasso without a bias penalty, and entries of the resultant precision matrix denote estimates of partial correlation for every pair of variables.

**Hardware details.** The experiments are run on a computer with AMD Ryzen Threadripper 3970X 32-Core Processor, two Nvidia Titan RTX GPU, and 188GB of RAM.

## H   Additional Experiments

In this section, we provide additional simulations to demonstrate Fair GLASSO behavior under hyperparameter tuning and violation of assumptions. Figures 4 and 5 of the attached document present Fair GLASSO performance as hyperparameters $\mu_1$ and $\mu_2$ vary when estimating a fair or unfair precision matrix, respectively. Observe that when the true precision matrix is unfair, increasing $\mu_2$ encourages a fairer estimate and thus increases the error. Moreover, smaller values of $\mu_2$ yield greater bias for the unfair setting in Figure 2 than the fair setting in Figure 4. While a larger $\mu_2$ decreases the bias in both settings, the effect is greater in Figure 5 for a true precision matrix that is unfair.

Moreover, we provide additional simulations in Figure 6 where the precision matrix varies in sparsity (AS1), the true precision matrix is rank-deficient (AS2), and the group sizes vary (AS4). Figure 6a shows the classical graphical lasso result, where as the precision matrix grows denser, estimation error suffers, particularly when the sparsity penalty weight $\mu_1$ is larger. In Figure 6b, we demonstrate the effects of rankness on estimation performance. Indeed, the use of $\epsilon > 0$ permits low-rank estimates, and we observe relatively robust estimation error for different values of $\epsilon$. Finally, Figure 6c shows that as the ratio between two groups becomes small, that is, the precision matrix becomes unfair due to imbalanced groups, error increases, particularly for larger $\mu_2$.

## I   Limitations and Future Work

This work focuses on group fairness via minimizing DP gap for Gaussian graphical models. While our proposed definition of DP for graphical models does not require Gaussianity, the theoretical

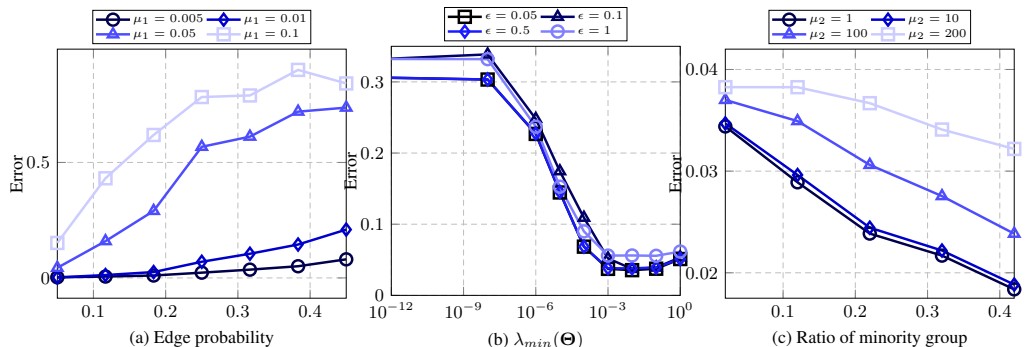

Figure 6: Performance in terms of the Frobenius error as different assumptions are violated. (a) Error as true precision matrix becomes denser. (b) Error as eigenvalues of true precision matrix grow close to 0. (c) Error as group sizes become increasingly imbalanced.

results and optimization algorithm require assuming both Gaussian random variables and use of our proposed metrics. Interesting and critical extensions of this work include more general graphical models and other ideas of fairness. Consider for instance the following adaptation of equalized odds (EO) [21],

$$\mathbb{P}\left[\boldsymbol{\Theta}^*_{ij} \mid [\boldsymbol{\Theta}_0]_{ij}, Z_{ia} = 1, Z_{ja} = 1\right] = \mathbb{P}\left[\boldsymbol{\Theta}^*_{ij} \mid [\boldsymbol{\Theta}_0]_{ij}, Z_{ia} = 1, Z_{jb} = 1\right] \ \forall a, b \in [g], \quad (72)$$

where $\boldsymbol{\Theta}^*$ denotes an estimate of the true precision $\boldsymbol{\Theta}_0$, and $\mathbf{Z}$ is the group membership indicator matrix. The comparison of fairness metrics for graphs warrants separate investigation since many fairness metrics in machine learning have not yet been adapted to the graph setting. Such an analysis of fairness metrics ought to be applied to tasks beyond graph learning and is thus out of scope of this paper. Moreover, while we considered discrete sensitive attributes, we also aim to promote equitable treatment for continuous sensitive traits.

Regarding the implementation of Fair GLASSO, Algoritm 1 can handle any function where the smooth and non-smooth components are separable. In principle, this includes non-convex functions, but the convergence result in Theorem 2 requires the smooth components to be strongly convex. Weaker results exist for convex functions [34], but no convergence guarantees are available for non-convex functions. In addition, Section 4.2 demonstrates the estimation of graphs with up to 1000 nodes. However, the computational complexity scales more than linearly with the number of optimization variables, rendering our approach potentially unsuitable for graphs with millions of nodes. Nonetheless, existing works can efficiently estimate very large graphical models, potentially under additional assumptions [71–73]. We can combine these approaches with our proposed fairness metrics, although we may lose our guarantee of convergence in Theorem 2.

Additional future directions include consideration of more general families of graphs. While we consider the underexplored case of signed, weighted graphs for fairness, our work is specific to graphical models encoding conditional dependence. We aim to consider more general interpretations of graphs with real-valued edges, including the novel extension to directed graphs.

