# OpenReview forum: "Fair GLASSO: Estimating Fair Graphical Models with Unbiased Statistical Behavior"
_NeurIPS.cc/2024/Conference — NeurIPS 2024 poster_

### Official Review · Reviewer_fwTq · 2024-07-07

**Soundness:** 3
**Presentation:** 3
**Contribution:** 4
**Rating:** 7
**Confidence:** 4

**Summary:**

The article explores the issue of achieving fairness in Gaussian Graphical Models (GGMs), particularly in the presence of biased data. Such biases can lead to unfair behavior in the models. To address this issue, the authors propose two bias metrics aimed at achieving statistical similarity across groups with different sensitive attributes. Based on these metrics, the authors introduce a regularized graphical lasso method called Fair GLASSO, designed to obtain sparse Gaussian precision matrices with unbiased statistical dependencies across different groups. The authors also propose an efficient proximal gradient algorithm to obtain these estimates and analyze the tradeoff between fairness and accuracy.

**Strengths:**

（1）The authors are the first to propose definitions of fairness and bias metrics applicable to graphical models.
（2）Through theoretical analysis and experiments, the effectiveness of Fair GLASSO in mitigating bias while maintaining model accuracy is demonstrated.
（3）The method is compared with various existing approaches, showcasing Fair GLASSO's advantages in reducing bias and improving accuracy

**Weaknesses:**

（1）The implementation of the algorithm relies on complex matrix operations, which may pose high computational costs in practical applications.
（2）Although the experiments used multiple datasets, the variety and scale of these datasets are still limited, possibly not fully representing all practical application scenarios.
（3）There are some formatting issues in the paper, such as unnumbered equations on page 23.

**Questions:**

(1) Can the authors provide a detailed explanation for the choice of these two bias metrics? Specifically, how do these metrics capture fairness better than other potential metrics?
(2) Can the authors provide a detailed breakdown of the computational complexity for each step of the Fair GLASSO algorithm?
(3) How does the proposed Fair GLASSO method perform on non-Gaussian data? Have the authors considered extensions or modifications to handle such cases?
(4) The method relies on certain assumptions (e.g., bounded spectrum, equal group sizes). How sensitive is the performance of Fair GLASSO to violations of these assumptions?
(5) What specific evaluation metrics were used to compare the performance of Fair GLASSO with other methods? Are these metrics the most appropriate for assessing both fairness and accuracy?
(6) Can the authors provide more details on the real-world datasets used, specifically the nature of the sensitive attributes and the relevance of fairness in those contexts?
(7) The paper contains some unnumbered equations, such as those on page 23. Can the authors clarify the reasons for this and provide the necessary numbering for easier reference?

**Limitations:**

(1) Although the paper discusses the efficiency of the Fair GLASSO algorithm, a more detailed analysis or discussion of the impact of complexity on practical applications would be better. For example, discussing potential scalability issues with larger datasets or in real-time applications could be very helpful.
(2) The paper relies on certain assumptions, such as bounded spectrum and equal group sizes. Including a sensitivity analysis for these assumptions would be beneficial.
(3) The paper could provide more concrete examples or case studies demonstrating the application of Fair GLASSO in real-world scenarios, beyond synthetic and small-scale examples.
(4) Discuss the ethical implications of using Fair GLASSO in these fields. Emphasize the importance of stakeholder engagement and continuous monitoring to ensure that the implementation does not inadvertently harm the groups it aims to protect.

---

> ### Author Rebuttal · Authors · 2024-08-07
>
> # Rebuttal for fwTq:
>
> We greatly appreciate your positive feedback along with your valuable questions and suggestions.
> We hope that our responses below maintain your positive assessment.
>
> > **Answer to Question 1.**
>
> Thank you for your question.
> For graph-based works, the predominant choice of bias metric is demographic parity (DP).
> Thus, we approach the nascent task of graphical model estimation with a familiar bias metric to verify our approach with established measurements.
> However, our formulation is suited to other bias metrics such as equalized odds (EO), defined in the global response.
> While both DP and EO are popular fairness definitions, we cannot compute EO for the true precision matrix since it is conditioned on the ground truth connections.
> As the crux of our theoretical results relies on measuring the bias in the true precision matrix, we choose DP as our bias metric.
>
> > **Answer to Question 2.**
>
> We present a brief analysis for bias metric $H\_\\mathrm{node}(\\mathbf{\\Theta})$, but a similar analysis holds for other penalties such as $H(\\mathbf{\\Theta})$.
> The first step of Algorithm 1 is a proximal gradient step.
> Computing the gradient requires an inverse $( \\mathbf{\\Theta} + \\epsilon \\mathbf{I} )^{-1}$ and product $\\mathbf{A} \\mathbf{\\Theta}\_{\\bar{\\mathcal{D}}}$, both incurring $\\mathcal{O}(p^3)$ operations.
> The gradient step and soft-thresholding enjoy entry-wise computations with complexities $\\mathcal{O}(p^2)$.
> The projection step onto the set of positive semidefinite matrices involves an eigendecomposition of $\\dot{\\mathbf{\\Theta}}^{(k)}$ with complexity $\\mathcal{O}(p^3)$.
> Finally, the step size update $t^{(k)}$ only requires scalar operations, and the accelerated update of $\\check{\\mathbf{\\Theta}}^{(k+1)}$ involves $\\mathcal{O}(p^2)$ operations, so they can be neglected.
>
> If accepted, we will include these additional details in the revised version of the manuscript to strengthen its clarity.
>
> > **Answer to Question 3.**
>
> In Table 2 of our manuscript, we estimate real-world networks from real data, such as the discrete graph signals of the School and Friendship social networks.
> Thus, we empirically observe satisfactory performance even when observations are non-Gaussian, as queried by the reviewer.
> However, you also ask an important question: can Fair GLASSO be extended to non-Gaussian graphical models?
> Gaussianity specifies the loss in the objective of Fair GLASSO, but we may consider other distributions in the optimization problem, such as the Ising negative log-likelihood.
> Such a substitution requires altering our theoretical results, and in future work we will explore the fairness-accuracy relationship for non-Gaussian distributions.
>
> > **Answer to Question 4.**
>
> We thank you for this question.
> If accepted, we will provide a detailed discussion on the assumptions for Fair GLASSO.
> Here, we share a brief summary.
>
> First, AS1 merely defines the cardinality of the support of the true precision matrix.
>
> For the spectrum of the true covariance matrix $\\mathbf{\\Sigma}\_0$, eigenvalues with finite magnitudes as in AS3 is reasonable for any practical setting.
> AS2 may be violated for a rank-deficient $\\mathbf{\\Sigma}\_0$.
> However, letting $\\epsilon>0$ addresses both theoretical and implementation concerns, where $\\mathbf{\\Sigma}\_0$ need not be positive definite for convergence or the error bound.
> As would be expected, the error bound becomes perturbed based on the magnitude of $\\epsilon>0$.
>
> The final assumption can be relaxed to require only that no group vanishes as $p\\rightarrow\\infty$.
> If a group vanishes, then edges cannot achieve perfect balance across all pairs of groups, which will result in a lower bound on the second term of the error upper bound.
>
> Moreover, we present additional simulations in the attached document illustrating how the assumptions affect performance.
>
> > **Answer to Question 5.**
>
> Our metrics for error and bias are described more thoroughly in Appendix G.
> We employ the normalized squared Frobenius error, a standard metric for network inference works.
> We follow similar intuition for bias, where we normalize measurements to compare biases across networks without being affected by changes in graph size or edge weights across graphs.
>
> > **Answer to Question 6.**
>
> In real-world social network analysis, common network characteristics such as homophily can lead to negative outcomes across gender in both social and academic settings [A4].
> Hence, the School, Friendship, and Co-authorship networks require scrutiny with respect to fairness, where gender is a critical consideration.
> For demonstrative purposes, we use publication type as the sensitive attribute for Co-authorship as an example of data with more than two groups.
> Additionally, biases in recommendation systems can reproduce and even exacerbate existing harmful stereotypes [A5].
> The MovieLens dataset, a common benchmark for fair graph machine learning, exemplifies our ability to form unbiased models from networks used for recommendation systems.
> If our work is accepted, we will include this relevant discussion in the final version.
>
> > **Answer to Question 7.**
>
> Thank you for asking about the reasons for our choice.
> We followed a common approach of numbering equations to which we refer, which serves to provide lighter notation.
> However, we agree with the reviewer that numbering all equations may better facilitate the understanding of our results.
> If the reviewer deems it necessary, we will gladly make this change upon acceptance of this work.
>
> We also appreciate your valuable suggestion regarding further discussion of ethical implications, and we will augment our broader impact discussion if accepted.

---

> > ### Comment · Reviewer_fwTq · 2024-08-13
> > **Fair GLASSO: Estimating Fair Graphical Models with Unbiased Statistical Behavior**
> >
> > I would like to thank the authors for their detailed responses, clarification, and additional results. Most of my questions are solved and I raise my score.

---

> > > ### Author Response · Authors · 2024-08-13
> > > **Thank you for your thorough review**
> > >
> > > We thank the reviewer again for your perceptive questions and comments. It is clear that you read our paper in detail, and we greatly appreciate your thoroughness.

---

### Official Review · Reviewer_V8H3 · 2024-07-07

**Soundness:** 3
**Presentation:** 3
**Contribution:** 3
**Rating:** 6
**Confidence:** 3

**Summary:**

The authors propose a method for estimating Gaussian graphical models that are fair. To do this, the methodology uses Fair GLASSO, a regularized loss for estimating the precision matrix of the GGM which is fair with respect to sensitive attributes. The authors also provide theoretical results regarding the asymptotic errors of the estimated precision matrix.

**Strengths:**

- Overall, the paper is well written.
- The methodology developed is sound and supported by theoretical results.
- The experimental results show that the derived method is effective in practice.

**Weaknesses:**

- Please define modularity and "partial correlations within and across groups" for completeness.
- Theorem 1 provides asymptotic results in terms of p, but it would be more interesting to derive results which are asymptotic in the number of data points n. Usually, the number of nodes are fixed and the number of datapoints increases.
- It would be interesting for authors to investigate how the results change when the assumptions are violated.
- Also, is n fixed even as p increases to infinity in the result above? In this case, won't the covariance matrix become uninvertible when p > n?
- The results should also include a pareto frontier between error and bias when the regularization parameters vary (but p and n are fixed)

**Questions:**

See the weaknesses section

**Limitations:**

The authors should elaborate on the limitations of the proposed methodology a bit more.

---

> ### Author Rebuttal · Authors · 2024-08-07
>
> # Rebuttal for V8H3:
>
> We thank you for your thorough review and detailed questions.
> We are grateful for your kind words regarding the strengths of our work.
>
> > **Answer to Weakness 1.**
>
> Group-wise modularity is computed as in reference [19] of the paper, that is,
> $$
> Q(\\mathbf{\\Theta}) =
> \\sum\_{a=1}^g \\frac{\\mathbf{z}\_a^\\top \\mathbf{\\Theta}\_{\\bar{\\mathcal{D}}} \\mathbf{z} }{2s} - \\sum\_{a=1}^g \\left(\\frac{\\mathbf{z}\_a^\\top \\mathbf{\\Theta}\_{\\bar{\\mathcal{D}}} \\mathbf{z} }{2s}\\right)^2,
> $$
> where $s$ denotes the number of nonzero entries in $\\mathbf{\\Theta}$.
> To estimate partial correlation, we apply graphical lasso without a bias penalty, and entries of the resultant precision matrix denote estimates of partial correlation for every pair of variables.
>
> We thank you for your comment.
> We agree that these definitions are necessary for completeness.
> We will add more detailed versions of these definitions in the revised paper.
>
> > **Answer to Weakness 2.**
>
> Your point is well taken, and we clarify the meaning of the theoretical result.
> The result in Theorem 1 holds with high probability as either $n$ or $p$ increase; it is the probability of the error bound holding that converges to 1 as $p$ increases to infinity.
> We understand that this presentation appears vague.
> Thanks to your question, we will state this fact more clearly in the final version if accepted.
>
> > **Answer to Weakness 3.**
>
> Thank you for this comment.
> For lack of space, we outline the effects of violating our assumptions here, but we will provide a comprehensive discussion in the updated paper if accepted.
> We also provide empirical demonstrations of the assumptions on estimation performance in the attached document, described in the global response.
>
> Observe that since AS1 has no restrictions on permissible values of $s$, AS1 merely defines the number of edges in the true graphical model.
>
> AS2 and AS3 bound the eigenvalues of the true covariance matrix $\\mathbf{\\Sigma}_0$.
> In realistic settings, these eigenvalues will have finite magnitudes as in AS3, but the covariance may indeed be rank deficient, violating AS2.
> Theoretically, this affects the log-determinant bound in equation (18) of our paper, as we cannot finitely bound the difference above with the reciprocal of the smallest eigenvalue.
> However, in such a case, we may instead consider Theorem 1 with $\\epsilon > 0$.
> We then obtain a similar error bound, albeit perturbed by the value of $\\epsilon$, that is, the magnitude of $\\epsilon$ increases with the number of zero-valued eigenvalues of $\\mathbf{\\Sigma}_0$.
>
> Finally, AS4 can be relaxed to require only that group sizes be asymptotically similar, so no group vanishes as $p\\rightarrow\\infty$.
> If a group vanishes, then the error bound will contain persistent terms corresponding to average edge magnitudes of the remaining groups, which cannot be balanced in connection across all groups.
>
> > **Answer to Weakness 4.**
>
> As in our response to your Question 2, we note that our theoretical results do not require that $n$ be fixed.
> Moreover, you are correct that with insufficient samples, the empirical covariance matrix may not be invertible.
> Indeed, a major advantage of graphical lasso is that it is suitable in a low sample regime.
> The sparsity penalty implements prior assumptions of parsimonious entries in the precision matrix to supplement inadequate information in the sample covariance matrix.
>
> > **Answer to Weakness 5.**
>
> We appreciate your valuable suggestion.
> We propose the following approach to show conditions on parameters $\\mu_1$ and $\\mu_2$ to guarantee Pareto optimality of Fair GLASSO.
> Say that we assume that our Fair GLASSO estimate $\\mathbf{\\Theta}^*$ has error $\\|\\mathbf{\\Theta}^* - \\mathbf{\\Theta}_0\\|_F^2 = \\delta$ for some $\\delta > 0$.
> For any feasible precision matrix $\\mathbf{\\Theta} \\in \\mathcal{M}$ such that $\\|\\mathbf{\\Theta} - \\mathbf{\\Theta}_0\\|_F^2 \\leq \\delta$, we wish to determine the selection of $\\mu_1$ and $\\mu_2$ that guarantees $H(\\mathbf{\\Theta}^*) < H(\\mathbf{\\Theta})$, that is, that we have a Pareto optimal estimate $\\mathbf{\\Theta}^*$.
> Our proposed approach is to exploit the optimality of $\\mathbf{\\Theta}^*$, which yields
> $$
> \\mu\_2 \\left( H(\\mathbf{\\Theta}) - H(\\mathbf{\\Theta}^*) \\right) \\geq \\mathrm{tr}(\\hat{\\mathbf{\\Sigma}}(\\mathbf{\\Theta}^*-\\mathbf{\\Theta})) + \\log \\det \\mathbf{\\Theta} - \\log \\det \\mathbf{\\Theta}^* + \\mu\_1 \\left( \\| \\mathbf{\\Theta}^*\_{\\bar{\\mathcal{D}}} \\|\_1 - \\| \\mathbf{\\Theta}\_{\\bar{\\mathcal{D}}} \\|\_1  \\right),
> $$
> from which we derive bounds on $\\mu_1$ and $\\mu_2$ to ensure that the right-hand side is positive.
> If our paper is accepted, we will provide the full version of the above Pareto optimality result, as requested by the reviewer.
>
> We appreciate your high standards regarding our paper.
> We will indeed elaborate on the limitations of our proposed work, the intuition of which your questions have helped us to develop.

---

### Official Review · Reviewer_QDSi · 2024-07-12

**Soundness:** 2
**Presentation:** 2
**Contribution:** 2
**Rating:** 4
**Confidence:** 4

**Summary:**

Traditional graphical models may reinforce existing biases present in the data. This paper introduces a novel approach to ensure that the learned graphical models are fair across different groups or demographics.

**Strengths:**

- The authors develop a penalty method that adds a fairness penalty to the GLASSO objective.
- As the added penalty is smooth, the authors develop a FISTA-type method to solve the optimization problem with an $\ell_1$ nonsmooth term. Objective includes Gaussian graphical model loss + fairness penalty term +  $\ell_1$ penalty term.
- The authors provide experiments on both real and synthetic datasets, demonstrating the efficacy of their methods.
- The paper is generally well-written and has a good synthetic empirical investigation.

**Weaknesses:**

- My main concern is that the proposed fair graphical model is essentially a joint graphical model widely studied in the existing literature. Please see below (Questions) for further clarification.
-  The proposed penalty functions are already studied in the existing graph fairness literature. Specifically, both fairness penalty functions $H(\Theta)$ and $H_{\text{node}}(\Theta)$ are from [15, 18], except that the authors replaced $| \cdot |$ with $( \cdot )^2$; for example, please refer to Eqs (1) and (2) in [18].
- Since $H (\Theta)$ and $H_{\text{node}}(\Theta)$ are smooth functions, the authors apply FISTA-type methods for graphical models. As FISTA and its variants are already explored in graphical models and their theoretical analyses are already provided [R2], Algorithm 1 simply applies to the smooth $f$ and nonsmooth $\ell_1$ term. Thus, the theoretical contribution of Theorem 2 is limited.
-  Estimation guarantees in Theorem 1 follow from existing literature on graphical model estimation, except that the authors need to handle $H(\Theta)$ and $H_{\text{node}}(\Theta)$ in the proof of Theorem 1. For example, the proof of trace difference, log-determinant difference, and sparsity penalties is identical to Theorem 1 of [32].
- In real experiments, the ground truth of the graph is used to generate the data, and then a fair graph is estimated. However, I am concerned that this might not be considered as real experiments since, in real scenarios, the precision matrix for graphical model estimation is indeed unavailable. In addition, it is not clear that these real datasets are suitable for Gaussian graphical models, as they should consist of binary numbers (e.g., interactions between people as nodes would be 0/1). This raises questions about the applicability of the method in real-world scenarios.

**Questions:**

-  How does the proposed method differ from the joint graphical lasso [R1, R3, R4, R5], which borrows strength across the groups  within graphical models that share certain characteristics, such as the locations or weights of nonzero edges? Specifically, under AS4 and since $Z$ is an indicator matrix, $H(\Theta)$ and $H_{\text{node}}(\Theta)$ are identical to the fused penalty term widely used in the joint estimation of joint graphical models; see for example [R4]. If you run joint graphical lasso [R1, R3, R4, R5], do they give similar results to yours?
- What is the difference from existing FISTA methods for graphical models; see for example [R2].  Can you clarify how the Hessian of $f$ in the FISTA section is positive definite? It would be good if you could clarify how the rate changes with the eigenvalue of $\nabla^2 f$.
-  What is the the key difference in bounding the trace difference, log-determinant difference, and sparsity penalties compared to Theorem 1 of [32] and existing joint estimation of graphical models [R1, R3, R4, R5]?
-  Can you clarify what are samples and nodes for your dataset? For example, what is the data matrix $X$ for the friendship experiment?
- Do these datasets resemble real-world applications of the proposed Gaussian graphical models, considering that i) they include a ground truth graph and ii) nodes of social networks have binary associated samples (e.g., 0/1 interactions between people)?





**Additional References**

- [R1] Pircalabelu, Eugen, and Gerda Claeskens. "Community-based group graphical lasso." Journal of Machine Learning Research 21.64 (2020): 1-32.
- [R2] Oh, Sang, Onkar Dalal, Kshitij Khare, and Bala Rajaratnam. "Optimization methods for sparse pseudo-likelihood graphical model selection." Advances in Neural Information Processing Systems 27 (2014).
- [R3] Guo, Jian, et al. "Joint estimation of multiple graphical models." Biometrika 98.1 (2011): 1-15.
-  [R4] Danaher, Patrick, Pei Wang, and Daniela M. Witten. "The joint graphical lasso for inverse covariance estimation across multiple classes." Journal of the Royal Statistical Society Series B: Statistical Methodology 76.2 (2014): 373-397.
- [R5] Ma, Jing, and George Michailidis. "Joint structural estimation of multiple graphical models." Journal of Machine Learning Research 17.166 (2016): 1-48.

---

> ### Author Rebuttal · Authors · 2024-08-07
>
> # Rebuttal for QDSi:
>
> We sincerely thank you for your review and your kind words.
> We appreciate your insightful comments and how you link our approach to existing works.
>
> > **Answer to Question 1.**
>
> Thank you for your interesting question.
> Your observation is key; similar to group lasso or joint graph inference methods, we optimize group-wise submatrices of the precision matrix, but our formulation differs subtly with important consequences for fairness.
>
> Joint graph inference methods such as [R1], [R4], and [R5] assume each graph lies on the same node set, where we may treat each submatrix of $\\mathbf{\\Theta}$ for every group pair as a different graph.
> Our topological fairness only requires that the support and signs of edges are balanced on average across all group pairs.
> However, joint graph inference yields similar sparsity patterns for all submatrices, which is far more restrictive than balancing edges in expectation.
> Moreover, the group lasso penalties in [R3], [R4], and [R5] can achieve fairness in support, but they ignore the signs of the edges, thus they cannot balance correlation biases.
> Thus, existing group lasso or joint graph learning approaches are either too restrictive or do not consider both kinds of graphical model bias.
> In contrast, our fairness metrics can balance structure in both sparsity patterns and signed edges.
>
> > **Answer to Question 2.**
>
> The main differences between our approach and other graphical model FISTA algorithms lie in (i) the fairness penalty in the objective function; (ii) the constraints for positive semidefinite precision matrices; and (iii) the type of convergence analysis.
> We first note that our Fair GLASSO algorithm and the work in [R2] have different objective functions and thus different gradient updates.
> Moreover, while [R2] solves an unconstrained optimization problem, we require projection onto the feasible set after soft-thresholding.
> On top of this, we guarantee convergence of the optimization variable, which is stronger than guaranteeing convergence of the objective function, as is done for [R2] and classical FISTA algorithms.
>
> To show that the Hessian of $f$ is positive semidefinite (PSD), here we supplement the proof of Theorem 2 (Appendix F).
> Note that $f$ can be split as $f = f\_1(\\mathbf{\\Theta}) + R\_H(\\mathbf{\\Theta})$, and recall that $\\mathbf{\\Theta}$ is PSD and $\\epsilon > 0$.
> Then, from (39), it follows that the Hessian of $f\_1(\\mathbf{\\Theta})$ is PSD since it is given by the Kronecker product of two PSD matrices.
> Next, note that the terms $R\_H(\\mathbf{\\Theta})$ promoting fairness are convex, so their Hessian is also PSD, rendering the Hessian of $f$ PSD.
>
> Finally, the convergence rate in Theorem 2 depends on the Lipschitz constant $L$ and the strong convexity constant $\\alpha$, respectively associated with the largest and smallest eigenvalue of the Hessian of $f$.
> Thus, it follows that increasing the smallest eigenvalues or decreasing the largest eigenvalues of the Hessian will improve the rate of convergence, and the converse is also true.
>
> Based on the feedback provided, we will highlight these relevant distinctions in the revised manuscript.
>
> > **Answer to Question 3.**
>
> You are correct that the proof is similar to that of the original result in [32].
> Since our proof is not limited by space, we aim to provide a self-contained result with enumerated steps for clarity, which can increase accessibility for audiences outside of statistics.
> We are also careful to identify the effects of bias mitigation.
> For example, the trace and sparsity differences do adhere to the proof in [32].
> However, the log-determinant difference in (18) of our paper deviates slightly from the original due to the additional bias term in the right-hand side of equation (13) in our manuscript versus equation (8) in [32].
> Moreover, our goal differs from [R1], [R3], [R4], and [R5], which show the effects on error when graphical lasso is modified to improve performance, while we formalize the tradeoff between accurate estimation versus unbiased solutions.
>
> > **Answer to Question 4.**
>
> For Karate club, School, Friendship, and Co-authorship, nodes represent individuals, while MovieLens nodes represent movies.
> Regarding samples, the Karate club dataset is the only one without real graph signals, so we generate synthetic Gaussian data given the ground truth network.
> MovieLens graph signals represent movie ratings.
> Edges in the Co-authorship network denote collaborations, and we use keyword frequencies as signals.
> Finally, for both the School and Friendship datasets, the edges represent pairwise student interactions over all time, and the graph signals are sums of interactions over windows of time, that is, the graph signals represent time-varying node degrees as interactions vary.
>
> Detailed descriptions of the real datasets can be found in the Appendix, and we will augment Table 3 to include the interpretation of nodes and signals.
>
> > **Answer to Question 5.**
>
> Your concern is warranted since real data simulations validate our approach in practical scenarios.
> Social network analysis is a common application of graphical models, but often no ground truth graph exists.
> However, we must confirm the effectiveness of Fair GLASSO.
> The Friendship and School datasets thus demonstrate our method for realistic applications, which we can verify with ground truth graphs.
>
> Critically, we emphasize that the Karate club dataset is the only real network for which we generate synthetic graph data.
> All other real-world datasets are each accompanied by a set of real graph signals.
> For example, the social networks possess real discrete graph signals, yet we still observe satisfactory performance in Table 2 of our paper.
> We hope this assuages your concern about the realism of our simulations.
>
> We sincerely thank you for your feedback, and we will refer to your thoughtful comments to clarify our method upon updating the manuscript.

---

> ### Comment · Reviewer_QDSi · 2024-08-14
> **Response to Rebuttal by Authors**
>
> I appreciate the authors' response. Below are my replies to your response.
>
> 1. In my opinion, the fair Glasso presented in this paper is indeed a variant of a graphical model with a group penalty, and further investigation with different choices of penalty formulations is needed. I also respectfully disagree with the statement that "group lasso penalties can achieve fairness in support, but they ignore the signs of the edges, thus they cannot balance correlation biases" without providing experimental or theoretical results. Indeed, there are variants of group penalties that show improvement in support recovery for sign-coherent groups. For example, (https://arxiv.org/pdf/1103.2697).
>
> 2. I believe an extensive discussion on the novelty of the method in the introduction, regarding both the penalty and the optimization method (FISTA for GLasso), is needed. For example, the phrases "For this purpose, we propose ..." and "we also propose a stronger alternative metric" in Lines 127 and 134 should be rephrased to explicitly mention that these penalties are from previous works such as [15, 18]. Further, the use of FISTA for smooth + $\ell_1$ objectives, as well as graphical lasso, is standard, and the convergence analysis in Theorem 2 follows from existing literature.
>
> 3. In my opinion, having no space limits does not allow us to repeat others' proofs. I recommend that the authors remove the identical parts (e.g., trace difference, log-determinant difference, and sparsity penalties) and frequently reference the modified parts of the estimation proof  (e.g., from Theorem 1 of [32]).
>
> 4. I still did not receive a response to my question, "What is the data matrix $X$ for the Karate Club, School, and Friendship datasets?" Is this data matrix generated using the ground truth graph, or is it available as real data for graph estimation? I don't think the statement "For both the School and Friendship datasets, the edges represent pairwise student interactions ..." answers my question. If I understand correctly, you are using this edge information to construct the ground truth graph, but it is not clear to me what $X$ in Eq. (4) represents for these datasets and how you constructed it. Also, I am uncertain whether these datasets (Karate Club, School, and Friendship datasets) are suitable for Gaussian graphical models, or whether another form of graph learning method should be applied. Do we have good literature on applying GLasso to these datasets, or should we consider using another type of graph learning?

---

> > ### Author Response · Authors · 2024-08-14
> > **Addressing additional comments from Reviewer QDSi**
> >
> > 1.  Your point is well taken; indeed modifications of group lasso penalties are plentiful, including those that consider signs of entries. However, as you rightly noted, these metrics show improvement in support recovery, but this is a different goal than promoting balanced connections. In particular, the cooperative-Lasso penalty promotes parsimonious estimates while accounting for sign, but it does not aim to mitigate differences of groups of entries. The fairness metrics for graphs take the difference of weighted sums of edges rather than promoting group-wise sparsity. Due to lack of space and time, we save a theoretical and empirical comparison of these penalties with topological DP for the revision of this paper, if accepted.
> > 2. Thank you for your suggestion. We will make these necessary changes regarding the novelty upon acceptance.
> > 3. We understand your point. If deemed necessary, we can certainly remove redundant derivations such as the sparsity and trace differences.
> > 4. We apologize if the explanation of the graph signals in our response seemed ambiguous. We observe student interactions over time, and each graph signal is a window of time. The signal value for one student at a given window is the number of interactions in which that student participated. Regarding the social network datasets, indeed these are typically equipped with discrete graph signals as is the case for the datasets shown in this work. Empirically, we observe satisfactory performance even when assuming Gaussianity. However, in future work, we aim to generalize fair graphical models beyond Gaussianity, such as Ising models, which do account for discrete signals.
> >
> > We thank you again for your detailed comments. We truly appreciate how invested you are in our work. We hope that you find these answers satisfactory.

---

### Official Review · Reviewer_pQPd · 2024-07-18

**Soundness:** 3
**Presentation:** 3
**Contribution:** 3
**Rating:** 5
**Confidence:** 4

**Summary:**

The paper introduces Fair GLASSO, a method for estimating Gaussian graphical models (GGMs) that addresses biases in data with respect to sensitive nodal attributes. The authors propose two bias metrics to promote fairness in statistical similarities across different groups, leading to the development of Fair GLASSO, a regularized graphical lasso approach. The paper also presents a proximal gradient algorithm for efficient estimation. Theoretical analysis shows the tradeoff between fairness and accuracy, and empirical results validate the effectiveness of the proposed method on both synthetic and real-world data.

**Strengths:**

1. Novel contribution in defining fairness for graphical models and proposing methods to estimate fair GGMs.
2. Thorough theoretical analysis including error bounds and convergence guarantees.
3. Comprehensive empirical evaluation on both synthetic and real-world datasets.
4. Proposed method can improve both fairness and accuracy in certain scenarios, especially when the underlying graph is fair but the data is biased.
5. Clear explanations and intuitive visualizations of concepts and results.

**Weaknesses:**

1. This work is limited to Gaussian graphical models and may not be able to generalize to other types of graphical models such as ising model or covariance model.
2. The fairness focus is mainly focusing on demographic parity and this may not be the optimal in relality. It would be better to explore other fairness definitions even though the authors claim that other definitions of group fairness can be similarly adapted.
3. Some of the real-world datasets use synthetic signals, which may limit the real-world applicability of those specific results.
4. The method introduces additional hyperparameters such as $\mu_1$ and $\mu_2$ that need to be tuned.

**Questions:**

1. How sensitive is the method to the choice of hyperparameters $\mu_1$ and $\mu_2$?
2. How would the approach extend to non-Gaussian graphical models?
3. How you considered other fairness metrics beyond demographic parity?
4. How does the computational complexity scale for very large graphs (e.g. millions of nodes)?

**Limitations:**

1. Only applicable to Gaussian graphical models.
2. Focuses solely on group fairness via demographic parity.
3. May not generalize well to more general graph structures.
4. Theoretical guarantees assume specific conditions on the true precision matrix and group sizes.
5. Limited exploration of the trade-offs between fairness and accuracy across different scenarios.

---

> ### Author Rebuttal · Authors · 2024-08-07
>
> # Rebuttal for pQPd:
>
> We are grateful for your positive review and clear questions.
> Indeed, your review helps us clarify the utility of our approach beyond conceptual discussions.
> We are glad to hear that you find our work novel, thorough, and comprehensive.
>
> > **Answer to Question 1.**
>
> Thank you for your question.
> Indeed, as with many graphical lasso modifications, we require selection of appropriate weights for additional penalties.
> Figures 1 and 2 of the attached document present Fair GLASSO performance as hyperparameters $\\mu_1$ and $\\mu_2$ vary when estimating a fair or unfair precision matrix, respectively.
> Observe that when the true precision matrix is unfair, increasing $\\mu_2$ encourages a fairer estimate and thus increases the error.
> Moreover, smaller values of $\\mu_2$ yield greater bias for the unfair setting in Figure 2 than the fair setting in Figure 1.
> While a larger $\\mu_2$ decreases the bias in both settings, the effect is greater in Figure 2 for a true precision matrix that is unfair.
> As hyperparameter tuning is a highly practical consideration, we will add these results to the final version if accepted.
>
> > **Answer to Question 2.**
>
> Your question allows us to clarify the flexibility of our bias metrics for graphical model estimation.
> Indeed, Gaussianity is widely used for its ubiquity and appealing statistical properties, yielding copious theoretical guarantees that we exploit for our results.
> However, note that Gaussianity only dictates the loss to be minimized, thus any optimization-based graph learning approach is amenable to our fairness penalties.
> Moreover, our FISTA algorithm is still applicable under other distributions as long as the associated loss is convex and differentiable, such as the negative log-likelihood of the Ising model that you mentioned.
>
> > **Answer to Question 3.**
>
> You raise an excellent point; notions of fairness may differ by application, thus consideration of other definitions is critical.
> Since fair graphical model estimation is a novel task, we follow the custom of existing graph fairness works that employ demographic parity (DP).
> However, the penalty for Fair GLASSO is flexible and allows for other bias metrics.
> Moreover, as long as the chosen fairness penalty is convex and differentiable, our convergent FISTA algorithm remains applicable.
> Consider for instance the following adaptation of equalized odds (EO),
> $$
> \\mathbb{P}[ \\Theta^*\_{ij} \~|\~ [\\mathbf{\\Theta}\_0]\_{ij},Z\_{ia}=1, Z\_{ja}=1 ] =
> \\mathbb{P}[ \\Theta^*\_{ij} \~|\~ [\\mathbf{\\Theta}\_0]\_{ij},Z\_{ia}=1, Z\_{jb}=1 ] ~\\forall a,b\\in[g],
> $$
> where $\\mathbf{\\Theta}^*$ denotes an estimate of the true precision matrix $\\mathbf{\\Theta}\_0$, and $\\mathbf{Z}$ is the group membership indicator matrix.
> Note that graphical EO is conditioned on the true precision matrix.
> Thus, we can measure biases in estimated precision matrices but not the true one.
> For this reason, we emphasize DP for group fairness since a measure of bias in the true precision matrix is critical to our theoretical interpretation of the fairness-accuracy tradeoff.
>
> > **Answer to Question 4.**
>
> While we show estimation of graphs on the order of 1000 nodes, the complexity indeed scales more than linearly with the number of optimization variables, thus our approach may not be viable for millions of nodes.
> However, existing works can efficiently estimate very large graphical models, potentially under additional assumptions [A1,A2,A3].
> We can combine these approaches with our proposed fairness metrics, although we may lose our guarantee of convergence.
>
> > **Answer to Weakness 3.**
>
> Finally, we wish to clarify a concern you expressed in Weakness 3.
> We stress that Karate club is the only real network for which we generate synthetic graph signals; all remaining real-world datasets possess data that are not synthetically generated.
> Indeed, as you noted, we wish to demonstrate the viability of Fair GLASSO in realistic settings, such as social networks with non-Gaussian data.
>
> We thank the reviewer for your questions. You pinpointed vital practical concerns that are crucial to address in order to continue exploring fair graph learning in realistic scenarios.

---

> > ### Comment · Reviewer_pQPd · 2024-08-12
> > **Concerns from Reviewer QDSi**
> >
> > Thank you for your detailed response. Most of my previous concerns are basically addressed. However, after taking a look at Reviewer QDSi's comments, I have some additional concerns as follows.
> >
> > (1) The main concern is about novelty. According to Reviewer QDSi's comment, the fairness penalty functions appear to be very similar to those in [15,18]. While you've clarified some differences from existing joint graphical models, the core idea still appears to be a variation on well-studied concepts. The modifications to balance sparsity patterns and signed edges, while interesting, do not seem to constitute a substantial leap forward in fair graph learning.
> >
> > (2) As for the technical contribution, the adjustments to the FISTA algorithm, including the constrained optimization and convergence guarantees, are incremental improvements rather than fundamental innovations. The theoretical analysis, while thorough, largely follows established approaches in graphical model estimation.
> >
> > (3) Your focus on demographic parity limits the broader applicability of the method. As noted in your response about equalized odds, adapting to other fairness metrics introduces additional complexities that are not fully addressed in the current work.
> >
> > Based on these new concerns, I believe this paper still needs improvement and I will drop my score from 6 to 5.

---

> > > ### Author Response · Authors · 2024-08-13
> > > **Addressing additional concerns from Reviewer pQPd**
> > >
> > > Thank you for bringing up your additional thoughts.
> > >
> > > (1)
> > > We completely understand your point that these previous works have contributed greatly to the analysis of fair network connectivity.
> > > The primary contribution of our work is the theoretical and empirical analysis of fair topologies of signed graphs, in particular, graphical models encoding conditional dependence.
> > > While [15,18] are seminal to fair graph signal processing, neither work can address the effects of fairness on graph topology when permitted to alter both sparsity and edge signs.
> > > In particular, the work in [15] aims to design graph filters for fair graph signal processing. An analogous task is performed in [18], which is restricted to balancing edges only by magnitude, while we further provide theoretical analysis for how fair graph learning affects the topology, a critical aspect of fairness for data science.
> > > Moreover, we note that the $\ell_2$ norm gives fairer outcomes even when edge sign is not considered, as group pairs are balanced overall as opposed to the $\ell_1$ norm which may favor balancing some pairs of groups over others.
> > >
> > > (2)
> > > Indeed, the advantages of FISTA for graphical model estimation is well known.
> > > While our analysis is inspired by existing works, we aim to show that not only are the convex fairness penalties amenable to efficient algorithms with well-understood performance guarantees, we also are able to provide a stronger guarantee that our algorithm converges with respect to the estimation variable, which is stronger than previous works' results on convergence of the objective function.
> > >
> > > (3)
> > > Fair methods aim to mitigate negative outcomes due to potentially hidden external influences such as stereotypes, thus we believe that thorough analysis is critical and aligned with the purpose of developing fair methods.
> > > We provide the first theoretical analysis of the effect of imposing fairness for graph estimation using a well-established notion of topological fairness.
> > > Indeed, the focus of this work is learning fair graphical models, while the comparison of fairness metrics for graphs warrants separate investigation since many fairness metrics in machine learning have not yet been adapted to the graph setting.
> > > Such an analysis of fairness metrics ought to be applied to tasks beyond graph learning and is thus out of scope of this paper.
> > >
> > > We thank you again for your comments, and we truly appreciate your high standards.

---

### Author Rebuttal · Authors · 2024-08-07

# Global

We would like to thank the reviewers for their quality comments and perceptive questions about our work.
Below we detail the main topics discussed both in the following responses and to be added to the revised paper should it be accepted.

We provide additional discussion of Fair GLASSO to clarify its flexibility and novelty.
Thanks to the questions by reviewer QDSi, we emphasize the novelty of our method and FISTA algorithm.
We also highlight the flexibility of Fair GLASSO; our choices of demographic parity and Gaussianity are convenient for our theoretical analysis but are flexible to alternatives, as inquired by reviewers pQPd and fwTq.
For example, we may instead measure bias in graphical models via Equality of Odds (EO), which we may define as
$$
\mathbb{P}[ \Theta^*\_{ij} \~|\~ [\mathbf{\Theta}\_0]\_{ij},Z\_{ia}=1, Z\_{ja}=1 ] =
\mathbb{P}[ \Theta^*\_{ij} \~|\~ [\mathbf{\Theta}\_0]\_{ij},Z\_{ia}=1, Z\_{jb}=1 ] ~\forall a,b\in[g],
$$
where $\mathbf{\Theta}^*$ denotes an estimate of the true precision matrix $\mathbf{\Theta}_0$, and $\mathbf{Z}$ is the group membership indicator matrix.

In the attached document, we provide additional simulations to demonstrate Fair GLASSO behavior under hyperparameter tuning and violation of assumptions.
Figures 1 and 2 of the attached document present Fair GLASSO performance as hyperparameters $\mu_1$ and $\mu_2$ vary when estimating a fair or unfair precision matrix, respectively.
Observe that when the true precision matrix is unfair, increasing $\mu_2$ encourages a fairer estimate and thus increases the error.
Moreover, smaller values of $\mu_2$ yield greater bias for the unfair setting in Figure 2 than the fair setting in Figure 1.
While a larger $\mu_2$ decreases the bias in both settings, the effect is greater in Figure 2 for a true precision matrix that is unfair.

We elaborate on the theoretical implications of violating the assumptions for Theorem 1, as requested by reviewers V8H3 and fwTq.
Moreover, we provide additional simulations in Figure 3 of the attached document where the precision matrix varies in sparsity (AS1), the true precision matrix is rank-deficient (AS2), and the group sizes vary (AS4).
Figure 3a shows the classical graphical lasso result, where as the precision matrix grows denser, estimation error suffers, particularly when the sparsity penalty weight $\mu_1$ is larger.
In Figure 3b, we demonstrate the effects of a low-rank ground truth precision matrix on estimation performance.
Indeed, the use of $\epsilon>0$ permits low-rank estimates, and we observe relatively robust error for different values of $\epsilon$.
Finally, Figure 3c shows that as the ratio between two groups becomes small, that is, the precision matrix becomes unfair due to imbalanced groups, error increases as we impose fairness, particularly for larger $\mu_2$.

We again thank the reviewers for taking the time to provide thorough evaluations of our submission.
We hope that you find our responses satisfactory.

References in Author Responses:
- [A1] C.-J. Hsieh, M. A. Sustik, I. S. Dhillon, P. K. Ravikumar, and R. Poldrack, "BIG & QUIC: Sparse Inverse Covariance Estimation for a Million Variables", in *Advances in Neural Information Processing Systems*, 2013.
- [A2] T. Yao, M. Wang, and G. I. Allen, "Fast and Accurate Graph Learning for Huge Data via Minipatch Ensembles", *arXiv preprint arXiv:2110.12067*, 2021.
- [A3] X. Wang, J. Ying, and D. Palomar, "Learning Large-Scale $MTP_2$ Gaussian Graphical Models via Bridge-Block Decomposition", in *Advances in Neural Information Processing Systems*, 2023.
- [A4] C. Avin, Z. Lotker, Y. Nahum, and D. Peleg, "Modeling and Analysis of Glass Ceiling and Power Inequality in Bi-populated Societies", in *International Conference and School on Network Science*, 2017.
- [A5] A.-A. Stoica, C. Riederer, and A. Chaintreau, "Algorithmic Glass Ceiling in Social Networks", in *International World Wide Web Conference*, 2018.

---

### Decision · Program_Chairs · 2024-09-25

**Decision:**

Accept (poster)

**Comment:**

The paper introduces Fair GLASSO, a method for estimating Gaussian graphical models (GGMs) that ensures fairness with respect to sensitive attributes in the data. The authors propose two bias metrics to promote fairness and develop a regularized graphical lasso approach, supported by theoretical analysis. The paper also presents an efficient proximal gradient algorithm for estimation and validates the method through empirical experiments on both synthetic and real-world datasets.

Pros:

Novel Contribution: Introduces fairness concepts in the context of graphical models, which is a relatively unexplored area.
Theoretical Rigor: Provides detailed theoretical analysis, including error bounds and convergence guarantees.
Empirical Validation: Comprehensive experiments on both synthetic and real-world datasets demonstrate the effectiveness of the method.
Clear Presentation: The paper is generally well-written and explains complex concepts in an accessible manner.

Cons:
Limited Novelty: The fairness penalty functions and the FISTA algorithm are extensions of existing methods, limiting the novelty.
Limited Generalization: The method is currently limited to Gaussian graphical models and may not generalize well to other types of graphical models.
Experimental Limitations: The experiments are conducted on a limited set of datasets, which may not fully represent the diversity of real-world scenarios.
Complexity Concerns: The computational complexity of the method may pose challenges in practical applications, particularly for large-scale problems.

Things that should be addressed in the final version of the paper:

Novelty: Concerns remain about the originality of the proposed fairness metrics and the incremental nature of the technical contributions.

Generalization and Flexibility: While the authors addressed concerns regarding the method’s generalizability to non-Gaussian models, the approach is still heavily reliant on specific assumptions and may not be flexible enough for broader applications.

Experimental Validation: Despite additional experiments provided in the rebuttal, there is still a need for more diverse and large-scale datasets to thoroughly validate the method.

The paper offers a meaningful contribution to the field of fair graphical model estimation, supported by solid theoretical work and empirical validation. The remaining concerns can be addressed by authors when preparing a camera ready submission.